# Immunomodulatory role of Keratin 76 in oral and gastric cancer

Inês Sequeira[1], Joana F. Neves[2], Dido Carrero[1], Qi Peng[3], Natalia Palasz[1], Kifayathullah Liakath-Ali[1,5], Graham M. Lord[2], Peter R. Morgan[4], Giovanna Lombardi[3] & Fiona M. Watt [1]

Keratin 76 (Krt76) is expressed in the differentiated epithelial layers of skin, oral cavity and squamous stomach. Krt76 downregulation in human oral squamous cell carcinomas (OSCC) correlates with poor prognosis. We show that genetic ablation of Krt76 in mice leads to spleen and lymph node enlargement, an increase in regulatory T cells (Tregs) and high levels of pro-inflammatory cytokines. Krt76$^{-/-}$ Tregs have increased suppressive ability correlated with increased CD39 and CD73 expression, while their effector T cells are less proliferative than controls. Loss of Krt76 increases carcinogen-induced tumours in tongue and squamous stomach. Carcinogenesis is further increased when Treg levels are elevated experimentally. The carcinogenesis response includes upregulation of pro-inflammatory cytokines and enhanced accumulation of Tregs in the tumour microenvironment. Tregs also accumulate in human OSCC exhibiting Krt76 loss. Our study highlights the role of epithelial cells in modulating carcinogenesis via communication with cells of the immune system.

[1] Centre for Stem Cells & Regenerative Medicine, King's College London, Guy's Hospital, Great Maze Pond, London SE1 9RT, UK. [2] Department of Experimental Immunobiology, King's College London, Guy's Hospital, Great Maze Pond, London SE1 9RT, UK. [3] Immunoregulation Laboratory, King's College London, Guy's Hospital, Great Maze Pond, London SE1 9RT, UK. [4] Department of Mucosal and Salivary Biology, King's College London, Guy's Hospital, Great Maze Pond, London SE1 9RT, UK. [5] Present address: Department of Molecular and Cellular Physiology and Howard Hughes Medical Institute, Stanford University Medical School, Stanford265 Campus DriveCA 94305-5453, USA. Correspondence and requests for materials should be addressed to F.M.W. (email: Fiona.Watt@kcl.ac.uk)

K eratins, the intermediate filament proteins of epithelial cells, are essential for normal tissue function, acting as a scaffold that enables cells to resist stress and damage[1]. Mutations that impair keratin assembly have been identified in a range of human skin disorders, typically leading to skin blistering or abnormal differentiation[2]. Recent studies have highlighted a novel role for keratins as regulators of inflammation and immunity in epithelia[3–8].

Krt76 is a type II intermediate filament protein expressed in the differentiating, non-proliferative layers of a subset of stratified epithelia in human and mouse[9]. Krt76 is the most significantly downregulated gene encoding a structural protein in human oral squamous cell carcinoma (OSCC) and correlates strongly with poor prognosis[10]. OSCC arises from the multilayered epithelial lining of the mouth and the lips. It involves mostly the tongue, but can also occur in the floor of the mouth, gingiva, lip, cheek and palate. Despite advances in treatment, the 5 year survival rate for OSCC remains stubbornly low, at 50–60%[11].

In patients, KRT76 is detected in 100% of normal gingivo-buccal epithelial biopsies, 44% of oral preneoplastic lesions and 35% of OSCC[10]. However, Krt76-null mice do not develop spontaneous OSCC, indicating that loss of Krt76 alone is not sufficient to induce tumours[10]. Nonetheless, genetic ablation of Krt76 in mice results in skin barrier defects, epidermal hyper-proliferation and inflammation[12,13], with mild hyperplasia and keratinisation of the buccal epithelium[10].

Here we have investigated the role of Krt76 in oral and stomach epithelial homoeostasis and the response of those tissues to the chemical carcinogen 4-nitroquinoline N-oxide (4NQO)[14], which mimics the carcinogenic effects of tobacco and alcohol ingestion[15]. We provide evidence for a previously unidentified role of Krt76 in regulating immunity in mice and demonstrate its importance in tumour progression.

## Results

**Keratin 76 is expressed in oral epithelia and stomach.** To analyse Krt76 expression and function, we used Krt76 mutant mice (Krt76[tm1a(KOMP)Wtsi]) generated by the Wellcome Sanger Institute Mouse Genetics Project[13,16,17] (Fig. 1a). As a result of splicing of the lacZ trapping element to Krt76 exon 2, homozygous mice do not express Krt76 (Krt76[−/−]). Heterozygous mice (Krt76[+/−]), expressing one copy of Krt76 and one copy of the lacZ reporter under the control of the endogenous promoter, were used to visualize Krt76 expression in the oral cavity and stomach. Krt76 was first expressed at embryonic day 17.5 (E17.5) in the tongue, palate and stomach (Fig. 1b, c) and expression continued in those locations throughout adulthood (Fig. 1e–i). Expression in the tongue occurred predominantly on the dorsal surface and lateral border, with fewer cells labelled in the ventral tongue (Fig. 1c–e). Krt76 was also strongly expressed in the palate (Fig. 1b, f). Expression was observed in the buccal mucosa but not in the outer lip, defining a clear boundary between the two epithelia (Fig. 1g). Krt76 expression was confined to the suprabasal layers in all oral epithelia (Fig. 1c–g, i).

Mouse stomach contains two well-defined areas: the non-glandular forestomach (also known as squamous stomach, which connects to the oesophagus) and the glandular stomach. Krt76 expression was confined to the suprabasal layers of the squamous stomach (Fig. 1h). Expression of Krt76 was further confirmed by antibody labelling (Fig. 1c, i) and qRT-PCR (Fig. 1j).

**Loss of Krt76 results in lymph node and spleen enlargement.** Although it was previously reported that loss of Krt76 results in hyperplasia of the buccal epithelium[10], we observed no histological abnormalities in the tongue or forestomach and no obvious

changes in epithelial proliferation in Krt76[−/−] mice, as assessed by EdU labelling (Fig. 2a–c). However, between the ages of 4 and 8 months all Krt76[−/−] mice spontaneously developed a large cyst (up to 17 mm in diameter) between the lower jaw and the fore-limbs (Fig. 2d–f). Each cyst was filled with fluid and was juxta-posed to the salivary gland (Fig. 2g). Immunofluorescence staining for B220 (CD45R) and CD3 revealed the presence of B and T cells, respectively, establishing that the cysts were enlarged submandibular lymph nodes (Fig. 2h, i). Flow cytometric analysis of the lymphocyte populations confirmed that the cyst cells were CD4[+] or CD8[+] mature T cells (Fig. 2j). Compared with the heterozygous controls (Krt76[+/−]), the lymph nodes in other body sites were also increased in size (Fig. 2k), even though Krt76 was not expressed in control lymph nodes (Supplementary Fig. 1a-b). Consistent with the increased size of the lymph nodes, there was a 2.2- to 3.9-fold increase in the absolute number of mesenteric, submandibular, axillary and inguinal lymph node cells in Krt76[−/−] mice compared with heterozygous littermate controls (Fig. 2l).

No changes were observed in thymus size or cellularity (Fig. 2m–o). However, Krt76[−/−] mice had enlarged spleens, with a 2.2-fold increase in the number of cells in the spleen (Fig. 2p–r). In addition, adult Krt76[−/−] mice consistently weighed less than control mice (Fig. 2s).

We conclude that loss of Krt76 results in splenomegaly and lymphadenopathy, indicative of systemic inflammation.

**Increased effector and regulatory T cells.** To dissect the inflammatory phenotype of Krt76[−/−] mice, flow cytometric analysis of immune cell populations was performed. The per-centage of total lymphocytes that were B cells (B220[+] TCRβ[-]) was increased in Krt76[−/−] mouse lymph nodes (Supplementary Fig. 1c-d) compared to heterozygous controls. The percentage of T cells (TCRβ[+] CD3[+] CD4[+]) that were effector T cells (CD4[+] CD44[high] CD62L[low]) was significantly increased in the spleen and lymph nodes (Fig. 3a). Effector T cells are known to play an important role in anti-tumour immunity[18]. There was also a significant increase in regulatory T cells (Tregs; TCRβ[+]CD4[+]CD3[+]Foxp3[+] [19]) in the lymph nodes and thymus of Krt76[−/−] mice, but not in the spleen (Fig. 3b). Tregs are potent anti-inflammatory cells that, among other functions, impede the anti-tumour immune response in a variety of cancers[20].

The changes in the levels of T effector cells, Tregs and B cells correlated with a striking upregulation of circulating IL-6, IL-10 and TNFα (Fig. 3c). The fluid inside the lymphoid cysts of Krt76[−/−] mice also had significantly elevated IL-6, IL-10 and TNFα (Fig. 3d). The levels of IFNγ, IL-2 and IL-4 were not altered (Fig. 3c). TNFα is known to stimulate Treg expansion[21].

To examine whether loss of Krt76 also resulted in local inflammation, we measured cytokine mRNA levels in the tongue and squamous stomach. This revealed a significant upregulation of TNFα in the tongue and IL-4 and TSLP in the squamous stomach (Fig. 3e). Consistent with these local increases in cytokine production, the total immune cell infiltrate (total CD45[+] cells) was significantly increased in the tongue and squamous stomach when compared to heterozygous controls (Fig. 3f, g). We also confirmed the previously reported increase in the skin inflammatory infiltrate[12,13] (Fig. 3f, g).

Taken together, these data demonstrate that loss of Krt76 results in local and systemic inflammation.

**Characterisation of Tregs and T effector cells.** To discover whether the functionality of Tregs and T effector cells differed in Krt76[−/−] and control mice we performed in vitro suppression assays (Fig. 4; Supplementary Fig. 2). Tregs were isolated from the

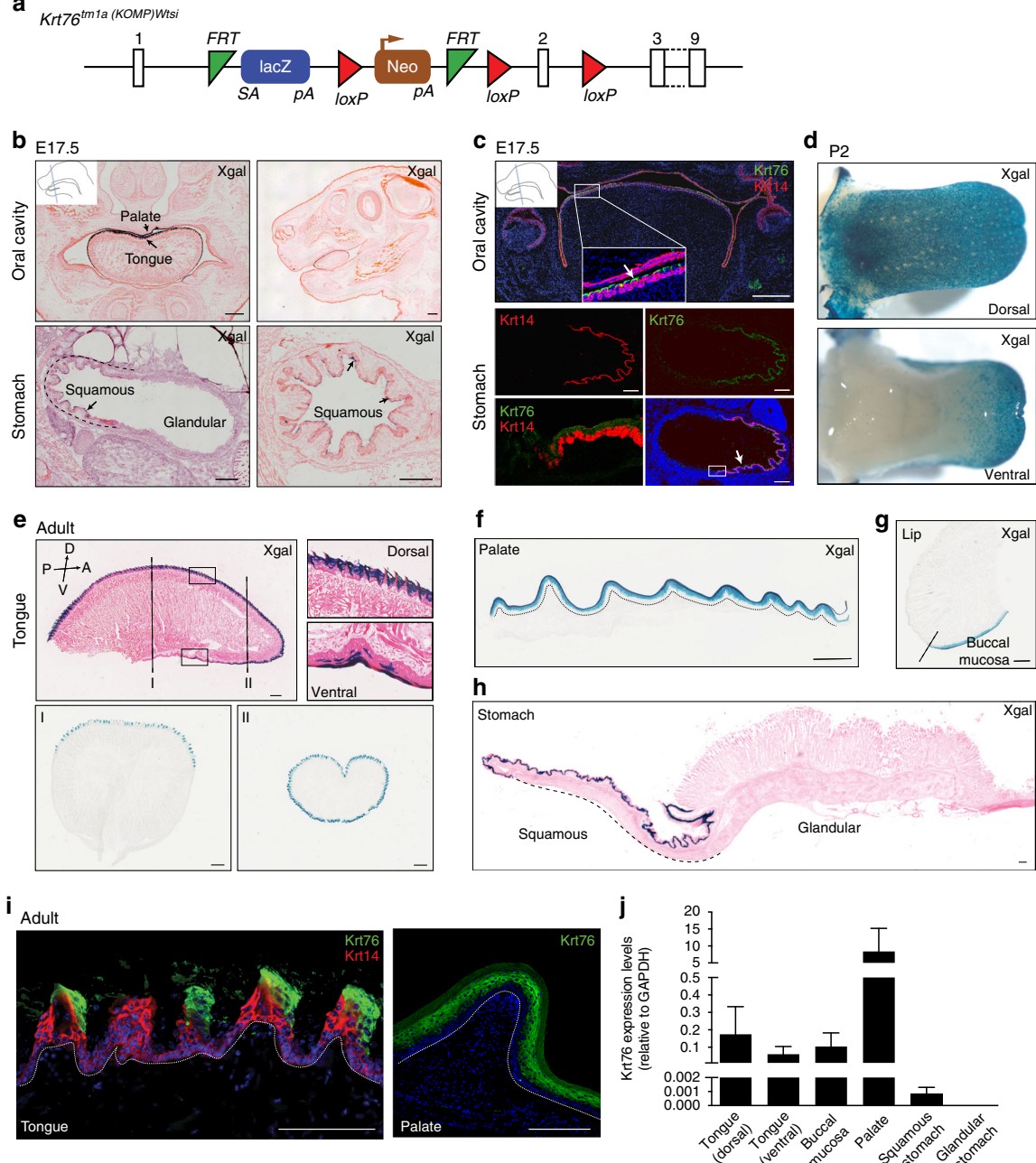

**Fig. 1** Keratin 76 is expressed in the oral epithelia and squamous stomach. **a** Krt76 knockout strategy. Krt76$^{-/-}$ mice were generated by disruption of the Krt76 gene via a knockout first allele targeting construct (reporter-tagged insertion with conditional potential). These animals have a splice acceptor-LacZ reporter gene integrated in the targeting gene, between exon 1 and 2, which allows tracing of gene expression whilst disrupting Krt76 protein expression. **b** X-gal staining (blue) of beta-galactosidase expressed under the control of the Krt76 promoter in the oral cavity and stomach (arrows) of Krt76$^{+/-}$ mouse embryos at E17.5. **c** Immunofluorescence labelling with anti-Krt76 (green) and anti-Krt14 (red) antibodies in the oral cavity and stomach of mouse embryos at E17.5. Bottom row: left hand panel is higher magnification view of boxed area in right hand panel. **d** Whole-mount X-gal staining of Krt76$^{+/-}$ reporter mice at post-natal day 2 (P2) shows Krt76 expression in the dorsal and lateral tongue, with partial expression in the ventral tongue. **e–h** X-gal staining (blue) of beta-galactosidase expressed under the control of the Krt76 promoter in tongue (**e**), palate (**f**), lip and buccal mucosa (**g**) and in stomach (**h**) of Krt76$^{+/-}$ adult mice. **h** Mouse stomach is subdivided into two major histologically distinct regions: the squamous stomach lined with a stratified squamous epithelium and the glandular stomach, separated by the limiting ridge from the stratified squamous epithelium of the squamous stomach. Krt76 expression is restricted to the squamous stomach region. **i** Immunofluorescence labelling with anti-Krt76 (green) and anti-Krt14 (red) antibodies of adult wild-type mouse tissues, confirming the specificity of both anti-Krt76 antibody and X-gal staining. Samples were counterstained with nuclear dye DAPI (4',6-diamidino-2-phenylindole). Dotted line delineates basement membrane. **j** Krt76 mRNA qRT-PCR analysis of adult tissues, relative to Gapdh (*n* = 3 mice, means ± s.e.m. are shown). Scale Bars = 500 μm (**b**, **c**), 100 μm (**e–i**)

spleen of control and Krt76$^{-/-}$ mice and co-cultured at different ratios with CFSE-labelled CD4$^+$CD25$^-$ responder T cells (Tresp), which include the T effector population. The suppression of proliferation was measured by flow cytometry (Fig. 4a, b). Krt76$^{-/-}$ Tregs inhibited Tresp proliferation more effectively than control Tregs, whether the Tresp were from control (Fig. 4b) or Krt76$^{-/-}$ mice (Supplementary Fig. 2a). In addition, the

proliferative activity of Krt76$^{-/-}$ Tresp in the absence of Tregs was lower than that of control Tresp (Fig. 4c).

The higher suppressive capacity of Krt76$^{-/-}$ Tregs correlated with increased capacity to inhibit expression of the pro-inflammatory cytokines IFNγ and IL17 and the anti-inflammatory cytokine IL-10, whether the Tresp cells were from control (Fig. 4d) or Krt76$^{-/-}$ mice (Supplementary Fig. 2b). Tregs from Krt76$^{-/-}$ mice expressed higher levels of the Treg

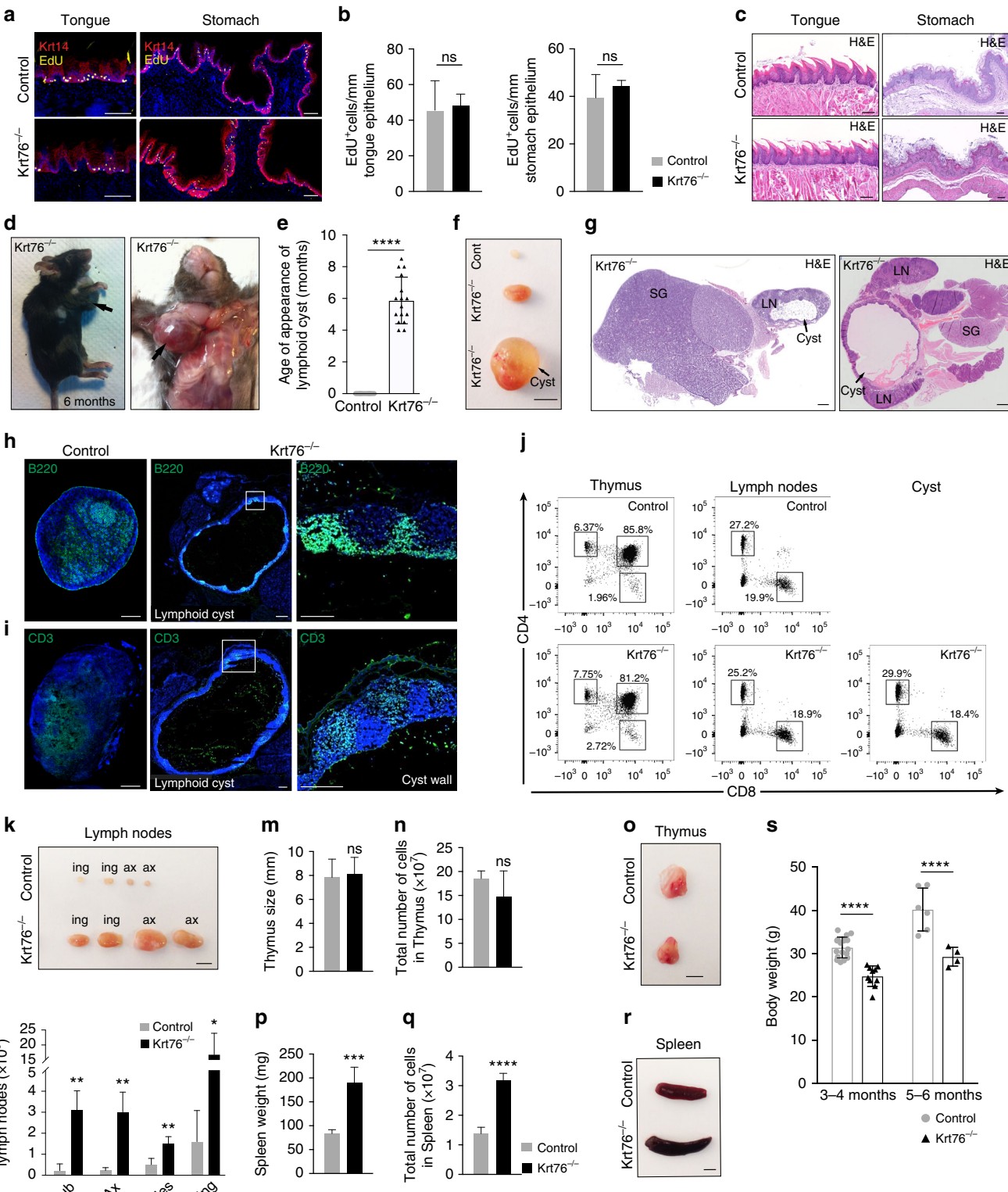

suppressive markers CD39 and CD73 compared to control Tregs (Fig. 4e, f). CD39$^+$ Tregs are known to suppress T cell proliferation and inflammatory cytokine production more efficiently than CD39 Tregs[22]. The differences between the properties of Tregs from Krt76$^{-/-}$ and control mice did not reflect a difference in Foxp3 expression levels (Fig. 4g).

We conclude that Tregs from Krt76$^{-/-}$ mice have enhanced suppressive function and Tresp are less proliferative compared with cells from control mice.

**Increased tumour incidence in Krt76$^{-/-}$ mice.** To determine whether the loss of Krt76 directly impacts tumour incidence in the oral cavity, we treated control (wild-type Krt76$^{+/+}$ and heterozygous Krt76$^{+/-}$) and Krt76$^{-/-}$ mice with a synthetic carcinogen 4NQO[14] (Fig. 5a) that mimics the carcinogenic effects of tobacco and alcohol ingestion[15]. In wild-type mice 4NQO induces carcinoma in the oral cavity and oesophagus, but not in the remainder of the digestive tract[23]. Mice received 100 μg/ml of 4NQO in the drinking water for 16 weeks and were monitored for the appearance and progression of lesions for a total of 28 weeks (Fig. 5a). Hyperplasia of the dorsal tongue was evident by 6 weeks and from 10 weeks mice had raised localised white lesions corresponding to dysplasias (Fig. 5a–c). From 16 weeks full OSCCs began to appear in all regions of the oral cavity (Fig. 5b, c) and by 22 weeks all mice had developed at least one tumour (Fig. 5f). The development of tumours was similar in male and female mice.

To explore the significance of the downregulation of Krt76 that occurs in human OSCCs[10], we monitored changes in the expression of Krt76 by LacZ labelling in 4NQO-treated Krt76$^{+/-}$ mice. Krt76 downregulation was first observed in hyperplastic oral epithelium (Fig. 5d; 13 out of 15 mice), consistent with the findings in human patients[10]. 10 out of 13 dysplasias and 4 out of 4 invasive SCC had focal or total loss of Krt76 (Fig. 5d, arrowed). Nevertheless, even when tumours developed within the same mouse, some lesions had lost Krt76 expression whereas others retained it (Fig. 5d, I and II). This was confirmed with immunostaining for Krt76; as expected (Fig. 1, Supplementary Fig. 3d), Krt76, when present, was co-expressed with differentiation markers such as Loricrin (Fig. 5e).

We next compared the incidence of oral cavity tumours in Krt76$^{+/+}$, Krt76$^{+/-}$ and Krt76$^{-/-}$ mice (Fig. 5f). There was no significant difference between Krt76$^{+/+}$ and Krt76$^{+/-}$ mice; however, Krt76$^{-/-}$ mice developed OSCC earlier. In Krt76$^{-/-}$ mice, the average onset of lesions was at 12 weeks after treatment ($n = 16$), compared to 17 weeks in control mice ($n = 14$ Krt76$^{+/+}$ and $n = 27$ Krt76$^{+/-}$, $p < 0.0001$). Furthermore, in Krt76$^{-/-}$ mice

the incidence was 100% by 14 weeks (Fig. 5f, $n = 16$). Many of the tumours had areas that were LacZ-negative, indicating downregulation of Krt76 or loss of Krt76-expressing cells (Fig. 5d).

None of the Krt76$^{+/+}$ or Krt76$^{+/-}$ control mice developed tumours in the stomach by 28 weeks from the start of 4NQO treatment (Fig. 5g). However, Krt76$^{-/-}$ mice started developing tumours in the squamous stomach by week 16 and by week 28 89% of the mice had a stomach tumour (Fig. 5g). No tumours or dysplasias were observed in the glandular stomach, regardless of genotype (Fig. 5h). As shown in Fig. 5h (arrow), there was extensive downregulation of Krt76 expression in stomach tumours.

We conclude that Krt76 ablation accelerates 4NQO-induced tumour progression in the tissues where Krt76 is expressed, namely the oral cavity and squamous stomach.

**Tongue and stomach epithelial integrity.** It has previously been reported that Krt76$^{-/-}$ epidermis exhibits reduced expression of tight junction proteins, Claudins, and an increase in the number of suprabasal layers[12,13] (Supplementary Fig. 3f). In contrast there was no reduction in Claudin1, Claudin3 or Claudin7 expression in Krt76$^{-/-}$ tongue or stomach epithelium (Supplementary Fig. 3f). There was no difference in the total thickness of tongue and stomach epithelium between control and Krt76$^{-/-}$ mice, nor in the suprabasal (Loricrin, Filaggrin or Involucrin-positive) layers (Supplementary Fig. 3d, e), although by Q-PCR a reduction in Involucrin was observed in Krt76$^{-/-}$ stomach (Supplementary Fig. 3f).

The epithelial integrity of the oral epithelia of Krt76$^{-/-}$ mice was assessed by a whole-mount Toluidine Blue dye penetration assay[24] and was found not to be defective (Supplementary Fig. 3b). Furthermore, following introduction of FITC-dextran by oral gavage there were no differences in the concentration of FITC-dextran in blood serum between control and Krt76$^{-/-}$ mice (Supplementary Fig. 3c). The previously reported delay in epidermal barrier formation[12] was confirmed (Supplementary Fig. 3a).

To examine whether there was increased penetration of commensal microorganisms that could lead to an activation of immune cells, we measured the bacterial load of the tongue and stomach epithelia by Gram staining and performed whole-mount fluorescence in situ hybridisation (FISH) with a universal bacterial probe (BacUni)[25]. Bacterial penetration was largely confined to the cornified layers of the tongue filiform papillae in both Krt76$^{-/-}$ and control mice, demonstrating that there was no

**Fig. 2** Loss of Krt76 leads to enlarged lymph nodes and spleen, without affecting tongue and stomach epithelial homoeostasis. Immunostaining (**a**) and quantification (**b**) of EdU-labelled cells per mm of tongue and squamous stomach epithelia ($n = 3$ mice/genotype, 2 sections/mouse and >6 fields quantified per section, means ± s.e.m. are shown). **c** Hematoxylin-eosin (H&E) stained sections. **d** Krt76$^{-/-}$ 6 month-old mouse with a neck cyst (arrowed). **e** Mean age (±s.e.m.) of onset of macroscopic neck lymphoid cysts in Krt76$^{-/-}$ ($n = 16$) and control mice ($n = 41$) (****$p \leq 0.0001$, unpaired $t$-test). **f** Macroscopic views of submandibular lymph nodes from control and Krt76$^{-/-}$ mice and cyst from Krt76$^{-/-}$ mouse. Pictures are representative of 9 mice/genotype. **g** Hematoxylin-eosin stained sections of neck lymphoid cyst inside the lymph nodes (LN) at early (left) and advanced (right) stages, juxtaposed to the salivary glands (SG). **h, i** Immunofluorescence staining of lymph nodes and lymphoid cyst sections with (**h**) anti-B220 (B cell marker) and (**i**) anti-CD3 (T cell marker) with DAPI counterstain. **j** Representative flow cytometry dot plots showing T cell populations from thymus (CD4$^+$ and CD8$^+$), submandibular lymph nodes (CD4$^+$ or CD8$^+$) and neck lymphoid cyst (CD4$^+$ or CD8$^+$, as the lymph nodes) from Krt76$^{-/-}$ and control mice, showing percentages of total live cells ($n = 3$ experiments, $n = 4$ mice/experiment/genotype). **k** Lymph nodes from control and Krt76$^{-/-}$ mice. Pictures are representative of 9 mice from each genotype. **l** Quantification of the total absolute number of cells in lymph nodes of control and Krt76$^{-/-}$ mice. **m, n** Thymus size (**m**) and thymus cell number (**n**) in control and Krt76$^{-/-}$ mice. **o** Representative photograph of thymus from control and Krt76$^{-/-}$ mice. **p, q** Spleen weight (**p**) and absolute spleen cell numbers (**q**) in control and Krt76$^{-/-}$ mice. **r** Representative photograph of spleens from control and Krt76$^{-/-}$ mice. **s** Total body weight of control ($n = 27$) and Krt76$^{-/-}$ ($n = 10$) mice. **l, m, n, p, q** mean ± s.e.m., *$p \leq 0.05$, **$p \leq 0.01$, ***$p \leq 0.001$, ****$p \leq 0.0001$, unpaired $t$-test, $n = 4$ mice/genotype, measured in duplicate and experiment repeated twice. Scale bars: 100 μm (**a, c, g–i**), 500 μm (**f, k, o, r**). Sub submandibular, Ax axillar, Mes mesenteric, Ing inguinal

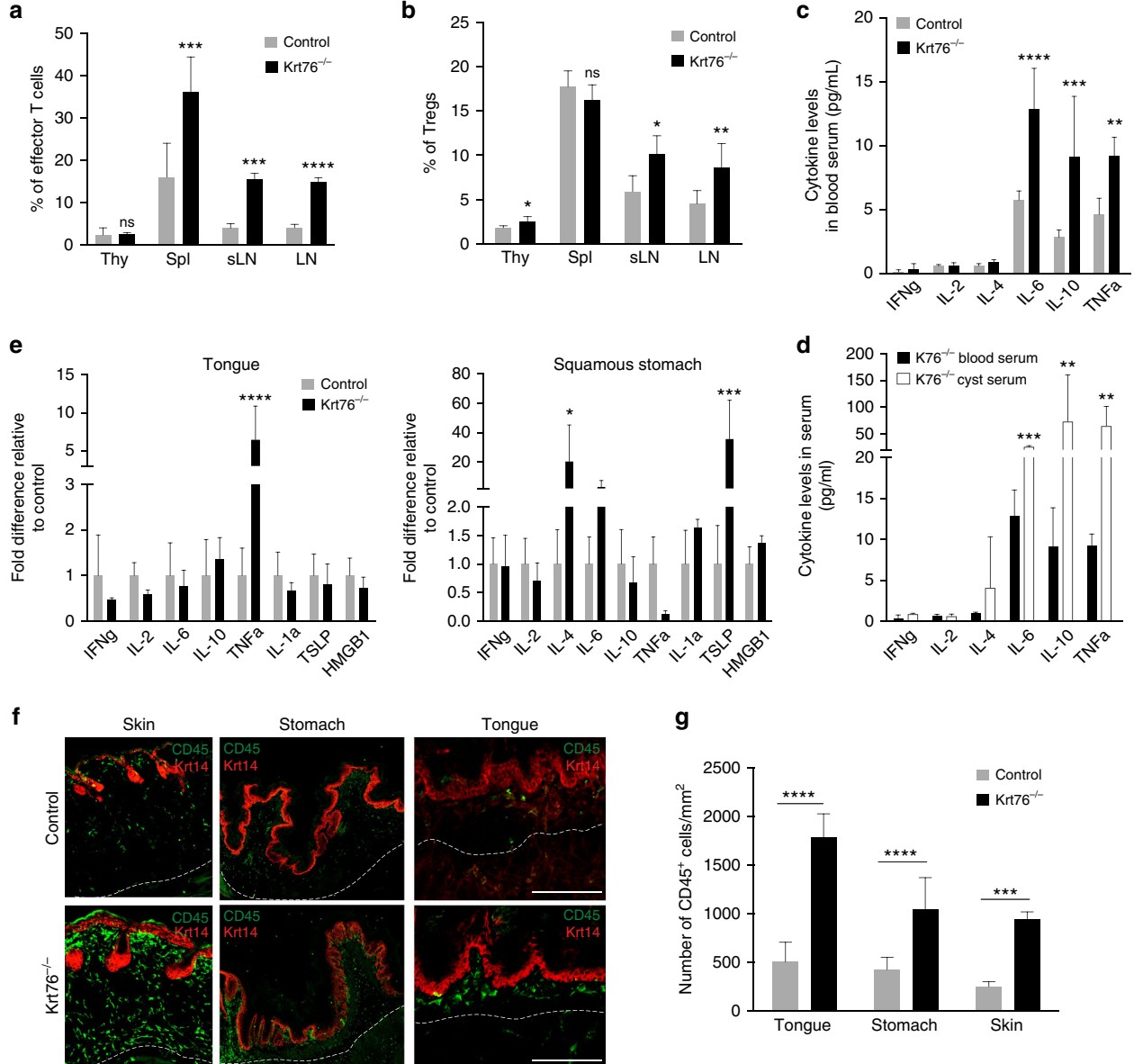

**Fig. 3** Loss of Krt76 results in local and circulating increase in cytokines and expansion of Tregs. **a, b** Summary of flow cytometric analysis of % effector T cells (CD44$^{high}$ CD62L$^{low}$) (**a**) and Foxp3$^+$ Tregs (**b**) in total TCRβ$^+$ CD3$^+$ CD4$^+$ T cells in thymus (Thy), spleen (Spl) and lymph nodes (LN; sLN, submandibular lymph nodes) from control and Krt76$^{-/-}$ mice ($n = 4$ mice/genotype, mean ± s.e.m., unpaired $t$-test). **c, d** Levels of cytokines in blood serum (**c**) and cyst fluid (**d**) of control and Krt76$^{-/-}$ mice assessed by CBA analysis ($n = 4$ mice per genotype; experiment repeated twice, mean ± s.e.m., unpaired $t$-test). The same blood serum measurements for Krt76$^{-/-}$ mice are shown in **c** and **d**. **e** Quantitative RT-PCR of cytokine mRNAs, relative to Gapdh, in tongue and squamous stomach epithelia ($n = 4$ mice/genotype, mean ± s.e.m. of biological and technical triplicates; unpaired $t$-test). **f, g** Representative images of CD45 (green) and Keratin14 (red) stained sections of skin, tongue and squamous stomach epithelia (**f**), and quantification of infiltrating CD45$^+$ leucocytes/mm$^2$ of stromal region (**g**). Stromal region corresponds to area between the epithelium and the white dotted line; $n = 3$ mice per condition, 2 sections per mouse and >4 microscopic views per mice, means ± s.e.m., ***$p \leq 0.001$, ****$p \leq 0.0001$, unpaired $t$-test). *$p \leq 0.05$; **$p \leq 0.01$; ***$p \leq 0.001$; ***$p \leq 0.001$; ns non-significant. Scale bars: 100 μm

significant difference in bacterial load or penetration (Supplementary Fig. 3g-h).

We conclude that loss of Krt76 did not compromise the integrity and barrier properties of tongue and stomach epithelia and that the increased tumour incidence in Krt76$^{-/-}$ mice is not linked to defective epithelial barrier formation.

**Local and systemic inflammatory response to carcinogen.** Since Krt76$^{-/-}$ mice exhibit local and systemic inflammation, we examined whether this was exacerbated by 4NQO treatment.

Tongue and squamous stomach were collected after 2 weeks of 4NQO treatment to assess whether the levels of inflammatory cytokines were altered. The levels of cytokines were increased upon 4NQO treatment of both Krt76$^{+/-}$ control and Krt76$^{-/-}$ mice compared to untreated mice (Fig. 6a compared to Fig. 3c). However, blood serum levels of IFNγ, IL-4, IL-6, IL-10 and TNFα were significantly higher in 4NQO-treated Krt76$^{-/-}$ than control mice (Fig. 6a).

In the tongue there were significantly more stromal CD45$^+$ cells in Krt76$^{-/-}$ versus control mice prior to 4NQO treatment (Fig. 3f, g), following 2 weeks of treatment and in 4NQO-induced

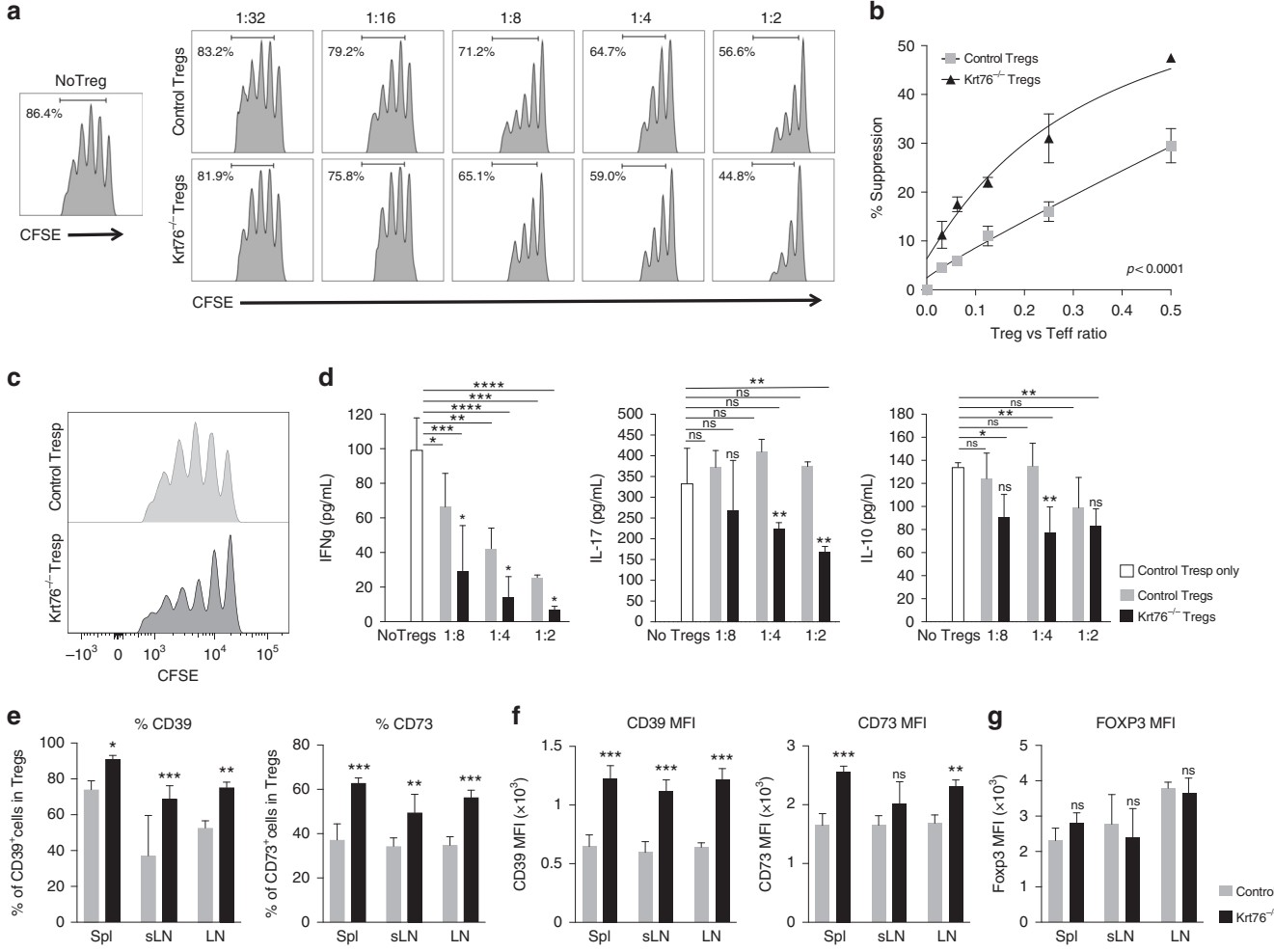

**Fig. 4** Phenotype and suppressive function of Krt76$^{-/-}$ Tregs. **a** Representative plots from a suppression assay involving different ratios of Tresp to Treg, showing CFSE profile quantified by flow cytometry after gating on CD4$^+$ cells. This illustrates dose-dependent suppression of wild-type CD4$^+$ CD25$^-$ Tresp proliferation in the presence of control or Krt76$^{-/-}$ Tregs. **b** Cumulative data showing % control Tresp suppression at each Tresp:Treg ratio in the presence of control or Krt76$^{-/-}$ Tregs. Mean ± s.e.m, two independent experiments, 2-way ANOVA, $p < 0.0001$, Non-linear regression (curve fit) of Treg suppression. **c** Representative plots of Tresp CFSE profiles after gating on CD4$^+$ cells, illustrating proliferation of Tresp from Krt76$^{-/-}$ and control mice in the absence of Tregs. **d** Levels of cytokines in cell culture medium for each Tresp:Treg ratio, assessed by CBA analysis (means ± s.e.m., multiple t-tests and two-way ANOVA, each culture condition in triplicate, measured in duplicate, experiment repeated twice). **e** Summary of flow cytometric analysis of % CD39$^+$ and CD73$^+$ Tregs from total Tregs (CD4$^+$ CD25$^+$ Foxp3$^+$) in spleen (Spl), lymph nodes (LN) and submandibular LN (sLN) from control and Krt76$^{-/-}$ mice. **f, g** Maximum intensity fluorescence (MFI) of CD39, CD73 (**f**) and Foxp3 (**g**) expression in CD4$^+$ CD25$^+$ Foxp3$^+$ Tregs ($n = 4$ mice/genotype, experiment repeated twice, mean ± s.e.m., unpaired t-test). *$p \leq 0.05$; **$p \leq 0.01$; ***$p \leq 0.001$; ****$p \leq 0.0001$; ns non-significant

hyperplasias (Fig. 6b, c). In Krt76$^{-/-}$ stomach, the number of local CD45$^+$ cells increased in the stroma adjacent to 4NQO-induced tumours (Fig. 6d).

Immunofluorescence labelling for Foxp3 revealed an increase of Tregs in both tongue and squamous stomach of Krt76$^{-/-}$ mice compared to control mice, whether normal or tumour-bearing tissue (Fig. 6e–h). The number of Tregs increased in hyperplasias and dysplasias of Krt76$^{-/-}$ tongue compared to controls and there was a marked accumulation of Tregs in tumour stroma (Fig. 6e, g, arrowheads, h). Furthermore, lesions in Krt76$^{-/-}$ tongue and stomach presented a concomitant decrease in effector T cells (Fig. 6i). Total CD4$^+$ cells were increased in the stomach but not in the tongue (Fig. 6i). Consistent with the findings in mice, there was an increase in stromal FoxP3$^+$ cells underlying Krt76-negative$^-$ regions of human OSCC (Fig. 7a) even when the differentiation marker Involucrin was still expressed (Fig. 7b).

Two cytokines that control Tregs are IL-18, which regulates Treg function[26,27], and IL-33, which is constitutively expressed in

barrier epithelial cells[28,29] and promotes Treg accumulation and maintenance in inflamed tissues[30]. Consistent with the accumulation of Tregs, IL-18 levels were significantly increased in 4NQO-treated (2 weeks) Krt76$^{-/-}$ compared to Krt76$^{+/-}$ tongue (Fig. 6j). In addition, IL-33 was increased in Krt76$^{-/-}$ tongue and squamous stomach after 4NQO treatment (Fig. 6j).

We conclude that the increased cancer susceptibility of Krt76$^{-/-}$ mice is correlated with an exacerbated systemic and inflammatory response to carcinogen, including the accumulation of Tregs in the tongue and squamous stomach.

**Increased Tregs correlate with accelerated tumour formation.** To examine whether targeting Tregs influences tumour onset in Krt76$^{-/-}$ mice, we developed mixed bone marrow (BM) chimeras by sublethally irradiating control and Krt76$^{-/-}$ mice and reconstituting them with BM from Depletion of Regulatory T Cell (DEREG) transgenic mice[31] (Fig. 8a, b). DEREG

mice express a diphtheria toxin receptor-enhanced green fluorescent protein (DTR-eGFP) fusion protein under control of the endogenous Foxp3 promoter, allowing both visualisation and diphtheria toxin-induced ablation of Foxp3[+] Tregs. Successful engraftment was verified by analysing the

percentage of Foxp3GFP[+] Tregs in the chimeras 6 weeks after reconstitution (Fig. 8c). As expected, there were more Foxp3GFP[+]Tregs in Krt76[−/−] than control mice (Fig. 8c).

We ablated donor Tregs by injecting diphtheria toxin (DT, or PBS as a control) for the first 5 weeks of the 4NQO carcinogenesis

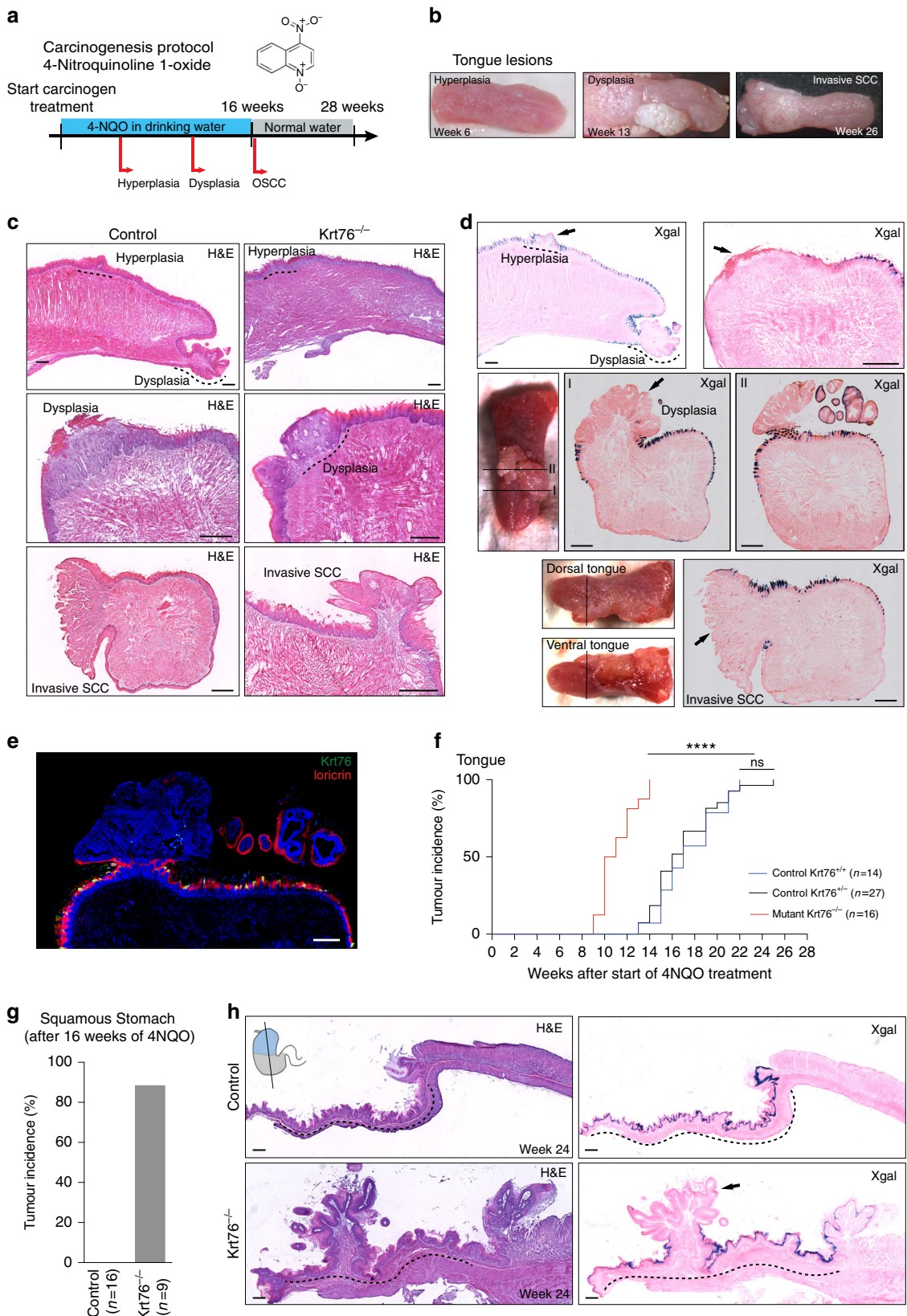

protocol (Fig. 8b). This time point was selected to coincide with the development of hyperplasia (Fig. 6h). After DT-injection for 5 weeks, donor Foxp3GFP$^+$ Tregs were significantly reduced in DEREG/Krt76$^{-/-}$ and control DEREG/Krt76$^{+/+}$ chimeric mice (Fig. 8c). However, when all the mice in each cohort had developed tumours and were subjected to end-point analysis, the total number of Tregs was significantly increased in DT-treated compared to PBS-treated DEREG/Krt76$^{-/-}$ mice (Fig. 8d). In control DEREG/Krt76$^{+/+}$ mice, the increase in lymph node Tregs following DT treatment was not statistically significant (Fig. 8d). However, in the spleen of DT-treated DEREG/Krt76$^{+/+}$ mice there was a significant increase in Tregs (Fig. 8e). Further analysis of Krt76$^{-/-}$ chimeras (Fig. 8f) showed that while, as expected, donor GFP$^+$ Tregs were reduced there was a significant increase in recipient GFP$^-$ Tregs, accounting for the increase in total Tregs.

Quantitation of 4NQO-induced carcinogenesis revealed that DT treatment accelerated tumour formation in both control and Krt76$^{-/-}$ mice (Fig. 8g). Thus higher levels of Foxp3$^+$ Tregs correlate with more rapid tumour development in both control and Krt76$^{-/-}$ mice. The major effect of Krt76 deletion on tumour susceptibility may therefore be via enhanced accumulation of Tregs.

## Discussion

Stratified, terminally differentiated epithelia, such as epidermis and oral epithelium, provide protective barriers against the environment, and their function depends on structural proteins, such as keratins. Until recently, keratins were mainly regarded as cytoskeletal scaffolds; however, there is an emerging role for keratins in the regulation of epidermal immunity[3–6,8,32,33]. For example, Krt1 is not only crucial to maintain skin integrity, but also regulates innate immunity by restricting IL-18 release from keratinocytes[5]. Krt17 acts as a regulator of skin immune responses and loss of Krt17 promotes reduced cell proliferation, leading to a delay in skin tumour onset[6]. Likewise, Krt16 regulates early inflammation and innate immunity in skin[32].

Krt76 is the most significantly downregulated gene encoding a structural protein in human OSCC[10,34] and downregulation correlates strongly with poor prognosis[10]. We have found that Krt76-null mice exhibit a marked inflammatory disease phenotype with systemic components: splenomegaly and lymphadenopathy (Fig. 2). This is correlated with a significant expansion of B cells (Supplementary Fig. 1c, d), effector T cells and Tregs (Fig. 3a, b), as well as an upregulation of inflammatory cytokines (Fig. 3c).

The Krt76$^{-/-}$ mouse serves as a model to explore the link between chronic inflammation and cancer, providing an opportunity to examine the impact of aberrant epithelial differentiation and consequent chronic inflammation on tumorigenesis. It is known that Krt76$^{-/-}$ mice do not develop spontaneous tumour lesions in the oral mucosa[10] or skin. However, we have shown that in response to 4NQO treatment Krt76$^{-/-}$ mice are more predisposed to developing tumours in those sites where Krt76 is normally expressed, namely the tongue and squamous stomach (Fig. 5f, g).

The enhanced sensitivity of Krt76$^{-/-}$ mice to carcinogenesis was not due to defective epithelial barrier formation. In addition, although some of the effects of Krt17 are attributable to its nuclear location[6,33], we failed to detect nuclear Krt76 by antibody labelling and Krt76 is not predicted to have a nuclear localisation signal[35]. Instead the cancer susceptibility of Krt76$^{-/-}$ mice correlated with a higher number of Tregs in secondary lymphoid organs and an enhanced accumulation of Tregs in the tongue and squamous stomach, which increased further in the tumour microenvironment (Fig. 6e–h). There was also a reduction in effector T cells in Krt76$^{-/-}$ mice in the tumour microenvironment. We conclude that keratins not only regulate inflammation and immunity in the skin (reviewed in[3]) but also in the oral cavity and squamous stomach.

Tregs were originally identified because of their ability to prevent organ-specific autoimmune disease by maintaining lymphocyte homoeostasis and regulating activated T cells[20]. However, there is emerging evidence that they play a major role in the tumour microenvironment[20,36] and contribute to tumour growth and progression by inhibiting the antitumor immune response[19,26,37]. Increased numbers of FoxP3$^+$/CD25$^+$ Tregs are observed in a subset of human OSCC and in the early stages of tumour progression in several mouse models[38]. While this has been linked to an immunosuppressive tumour microenvironment[39–42], there is still controversy about whether tumours with increased Tregs have a better or worse prognosis[43].

Krt76$^{-/-}$ mice exhibited deregulated cytokine expression. Anti-tumour function of dendritic cells is suppressed through IL-10[44] and we found IL-10 to be upregulated in Krt76$^{-/-}$ mice, particularly after 4NQO treatment. Increased levels of IL-10 are also reported in patients with OSCC[45]. However, IL-6, which is a key cytokine in encouraging cancer cell proliferation[46], was also upregulated in Krt76$^{-/-}$ mice. After carcinogen treatment, we observed an increase in serum IFNγ (Fig. 6a), consistent with the fact that IL-18, which is also upregulated (Fig. 6j), stimulates IFNγ production[47]. Krt76$^{-/-}$ effector T cells produced more IFNγ than control effector T cells in culture (cf. Figure 4d and Supplementary Fig. 2b).

In addition to being more abundant, Tregs from Krt76$^{-/-}$ mice had enhanced suppressive function, which correlated with higher levels of CD39 and CD73 expression (Fig. 4). CD39 is expressed on human and murine Tregs and is known to mediate immune T cell suppression by the downstream production of adenosine[48]. CD39$^+$ Tregs suppress T cell proliferation more efficiently than CD39$^-$ Tregs[22]. The coordinated expression of CD39 and CD73 on Tregs in Krt76$^{-/-}$ mice can explain their enhanced suppressive function. Tregs from CD39$^{-/-}$ mice have impaired suppressive function in vitro and fail to block transplant rejection in vivo[48]. Furthermore, Treg expression of CD39 and

---

**Fig. 5** Tongue and squamous stomach tumour incidence in Krt76$^{-/-}$ mice. **a** Schematic of 4NQO tumorigenesis protocol. **b** Representative macroscopic views of each stage of tongue tumour development. **c, d** Representative images of hematoxylin & eosin staining (H&E) (**c**) and X-gal staining (blue) (**d**) and respective macroscopic views of the tongue of control ($n = 41$ mice) and Krt76$^{-/-}$ mice ($n = 16$ mice). X-gal staining is used to visualize Krt76 expression. **e** Immunostaining for Krt76 (green) and Loricrin (red) (terminal differentiation marker) in a tongue section bearing two tumours, one of which expresses Krt76. **f** Tumour incidence in Krt76$^{+/+}$ ($n = 14$, median = 17 weeks), Krt76$^{+/-}$ ($n = 27$, median = 16 weeks) and Krt76$^{-/-}$ mice ($n = 16$ mice, median = 10.5 weeks) (no significant difference between the wild-type Krt76$^{+/+}$ and the heterozygous Krt76$^{+/-}$ controls; ****$p < 0.0001$ for Krt76$^{-/-}$ when compared to both controls; one-way ANOVA, Mantel–Cox test and Grehan–Breslow–Wilcoxon test). **g** Incidence of squamous stomach tumours in control (Krt76$^{+/+}$ and Krt76$^{+/-}$) ($n = 16$) and Krt76$^{-/-}$ mice ($n = 9$) harvested 16–28 weeks after the initiation of 4NQO treatment. **h** Representative images of H&E and X-gal staining (blue) of tumours in the squamous stomach. X-gal staining is used to visualize Krt76 expression. Dotted lines delineate the squamous stomach area. Scale bars: 100 μm

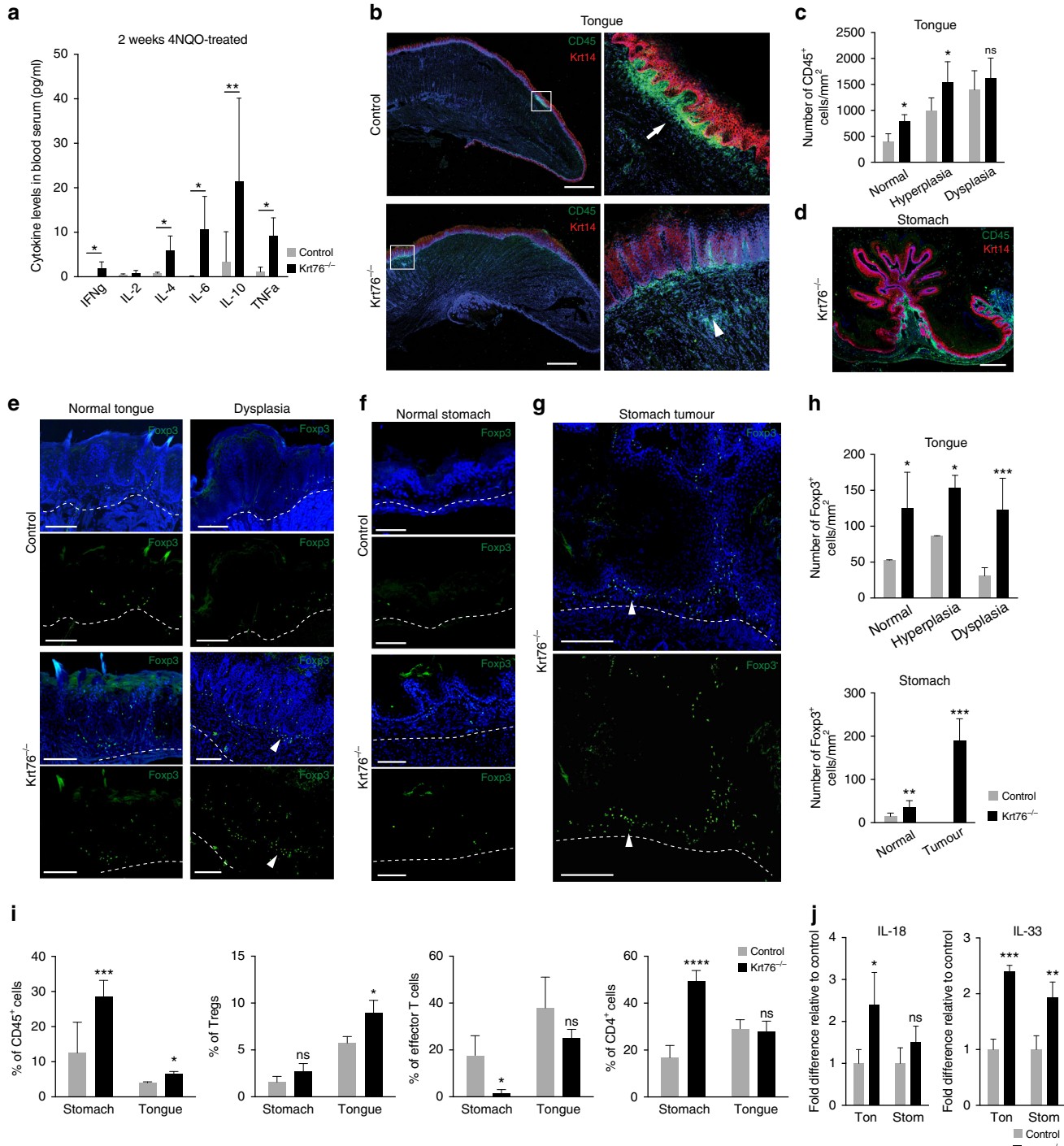

**Fig. 6** Infiltration of immune cell populations in tumour stroma. **a** Levels of cytokines in blood serum of control and Krt76$^{-/-}$ mice after 2 weeks of 4NQO treatment, assessed by CBA analysis (means ± s.e.m., unpaired *t*-test, 4 mice per genotype, measured in duplicate, experiment repeated twice). **b** Representative images of CD45 and Keratin 14 immunostaining in tongue lesions treated with 4NQO. Arrow and arrowhead indicate local accumulations of CD45$^+$ cells. **c** Quantification of infiltrating CD45$^+$ leucocytes in 4NQO-treated tongue ($n = 3$–8 mice/genotype, >4 microscopic views per region, means ± s.e.m., unpaired *t*-test). **d** Representative image of CD45 and Keratin 14 expression in stomach tumour. **e** Representative images of Foxp3$^+$ Tregs in 4NQO-treated normal tongue (i.e., prior to development of hyperplasia) and dysplasias in control and Krt76$^{-/-}$ mice. **f**, **g** Representative images of immunostaining of Foxp3$^+$ Tregs in 4NQO-treated normal stomach (**f**) and tumours (**g**). **e**–**g** Dashed lines denote epithelial-stromal boundary; arrowheads denote FoxP3$^+$ cells. **h** Quantification of infiltrating Foxp3$^+$ Tregs in 4NQO-treated tongue and squamous stomach ($n = 3$–8 mice per condition and per genotype, >4 microscopic views per region, means ± s.e.m., unpaired *t*-test). **i** Summary of flow cytometric analysis of % CD45$^+$, CD4$^+$ or effector T cells (CD4$^+$ CD44$^+$ CD62L$^{low}$) cells in total live cells, and Foxp3$^+$ Tregs in total CD4$^+$ CD25$^+$ T cells, in tongue and stomach epithelia after 4NQO treatment of control and Krt76$^{-/-}$ mice ($n = 4$ mice/genotype, means ± s.e.m., unpaired *t*-test). **j** Quantitative RT-PCR of IL-18 and IL-33 mRNAs in 4NQO-treated tongue and squamous stomach, normalised to glyceraldehyde 3-phosphate dehydrogenase ($n = 4$ mice/genotype, means ± s.e.m. of biological and technical triplicates; unpaired *t*-test). *$p \leq 0.05$, **$p \leq 0.01$, ***$p \leq 0.001$; ****$p \leq 0.0001$; ns non-significant. Scale bars: 100 μm

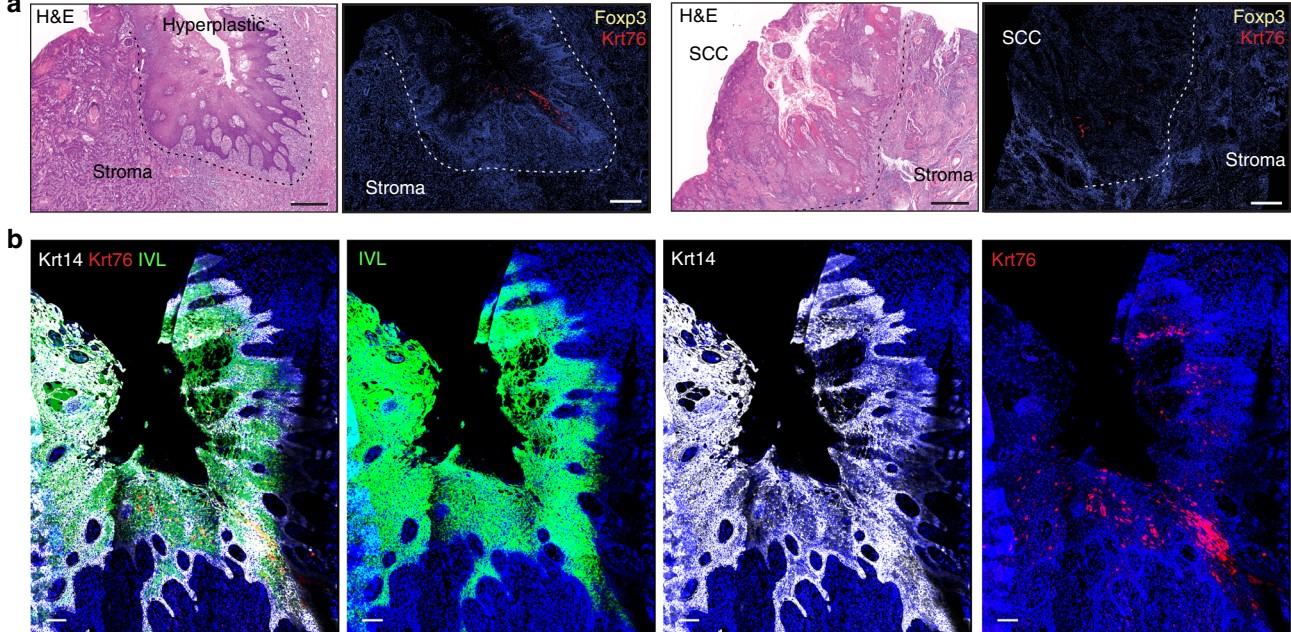

**Fig. 7** Krt76 expression and Foxp3[+] Treg infiltration in human OSCC. **a** Representative images of H&E staining and immunostaining for Krt76 and Foxp3[+] Tregs in human OSCCs. **b** Representative images of immunostaining of Krt76, keratin 14 (Krt14) and involucrin (IVL) with nuclear DAPI counterstain in human OSCC. Scale bars: 100 μm

CD73 is greater in human HNSCC than in healthy tissue[49], characterised by increased adenosine-mediated suppression of effector T cells[48,49].

In support of a positive role for Treg in 4NQO carcinogenesis, we found that partial depletion of donor Tregs in chimeric mice led to an increase in total Tregs and a corresponding acceleration of tumour formation, both in Krt76[−/−] and control mice. We propose that upon loss of Krt76 the increase in the immuno-suppressive Treg infiltrate leads to a failure in anti-tumour immunity linked to exaggerated suppression of anti-tumour-associated antigen-reactive lymphocytes. How loss of Krt76 exerts its effects requires further investigation; however, mechanisms involving inflammasome activation[50] or epithelial production of danger-associated molecular pattern (DAMP) proteins[51] are possibilities.

The inability to resolve chronic inflammation is considered one of the initial triggers of carcinogenesis[52], while immunosuppression is a crucial tumour immune-evasion mechanism and the main obstacle to successful tumour immunotherapy[20]. Our study highlights the importance of keratins as immunomodulators and the potential significance of highly suppressive CD39[+] Tregs in OSCC, making them potentially attractive targets for new cancer therapies. The Krt76[−/−] mouse provides a paradigm for under-standing how the differentiated epithelial layers contribute to oral carcinogenesis by provoking both local and systemic immune responses[43].

## Methods

**Animal procedures.** All animal procedures were subject to institutional ethical review and performed under the terms of a UK Home Office license. KRT76[−/−] (Krt76[tm1a(KOMP)Wtsi]) mice were obtained from the Wellcome Sanger Institute Mouse Genetics Project[13,16,17]. Heterozygous Krt76[+/−] mice were crossed to obtain knockout (Krt76[−/−]) and littermate control (heterozygous Krt76[+/−] or wild-type Krt76[+/+]) mice. DEREG mice[31] were kindly provided by Caetano Reis e Sousa (The Francis Crick Institute). Mice were maintained on the C57Bl/6 N genetic background.

In some experiments, mice received a dose of 500 μg EdU (5-ethynyl-2′-deoxyuridine, Invitrogen) in PBS intraperitoneally 2 h before tissues were harvested to assess proliferation. Tregs were depleted by intraperitoneal injection of 20 ng

diphtheria toxin (DTX; Sigma) per g mouse (as in Fig. 8b), once a week for 5 weeks[53]. Grafting efficiency was confirmed by analysing Foxp3 GFP[+] cells by flow cytometry at indicated time-points. Sample sizes were determined on the basis of prior power calculations.

**Human samples.** Human tissues were obtained from the Guy's & St Thomas' NHS Foundation Trust research biobanks, which are licensed by the Human Tissue Authority (licence number 12121).

**BM reconstitution.** In BM transplantation experiments (Fig. 8), 8- to 16-week-old male and female Krt76[−/−] and control littermate recipients (Krt76[+/−]) were treated with acidified water at least 10 days before irradiation. Statistical power was calculated using the resource equation and animals were randomly assigned to treatment groups. Allogenic BM transplants were performed 24 h after total body irradiation (two times 5.5 Gy, separated by 3 h). Donor BM was isolated from the tibia and femur of male mice. BM reconstitution was performed by intravenous injection of $2 \times 10^6$ BM cells in 200 μl PBS. Chimerism was confirmed by analysing Foxp3GFP[+] cells in the blood by flow cytometry.

**4NQO carcinogenesis.** 4NQO (Sigma, diluted to 100 μg/ml) was administered in the drinking water and fed to mice as the sole source of drinking water during the carcinogen-treatment period. 4NQO-containing water was prepared and changed once a week for 16 weeks. After that period, mice were given normal drinking water. During the experiments, the mice were maintained with regular mouse chow and water (with or without 4NQO) ad libitum. Once a week 4NQO-treated mice were sedated with inhaled isoflurane and the oral cavities were screened for lesions (hyperplasias, dysplasias and SCCs).

**Flow cytometry.** Lymph nodes, thymus and spleen were harvested from treated mice, mechanically disrupted, passed through a 40 μm cell strainer (BD Falcon) and rinsed with PBS to remove debris. Red blood cells were lysed using Ammonium–Chloride–Potassium Lysing Buffer. Cell number was determined using a Scepter Cell Counter (Merck Millipore). Tongue and stomach tumours were dissociated using Tumour Dissociation Kit (Miltenyi Biotec) according to the manufacturer's recommendations.

Single cells were labelled according to standard procedures. Briefly, single-cell suspensions were washed once in Stain Buffer and resuspended at $1 \times 10^6$ cells/ml. Nonspecific staining was blocked with Fc block (CD16/CD32, clone 93) prior to cell surface staining with the following antibodies: CD3ε PECy7 (145-2C11), CD3ε FITC (145-2C11), CD4 PerP.Cy5.5 (RM4-5), CD4 PE (RM4-5), CD8α PE (53–6.7), CD8α FITC (53–6.7), CD8 BV605 (53–6.7), TCRβ APC (H57-597), B220 APCCy7 (RA3-6B2), CD25 APCe780 (PC61.5), CD62L PerCPCy5.5 (MEL-14), CD73 FITC

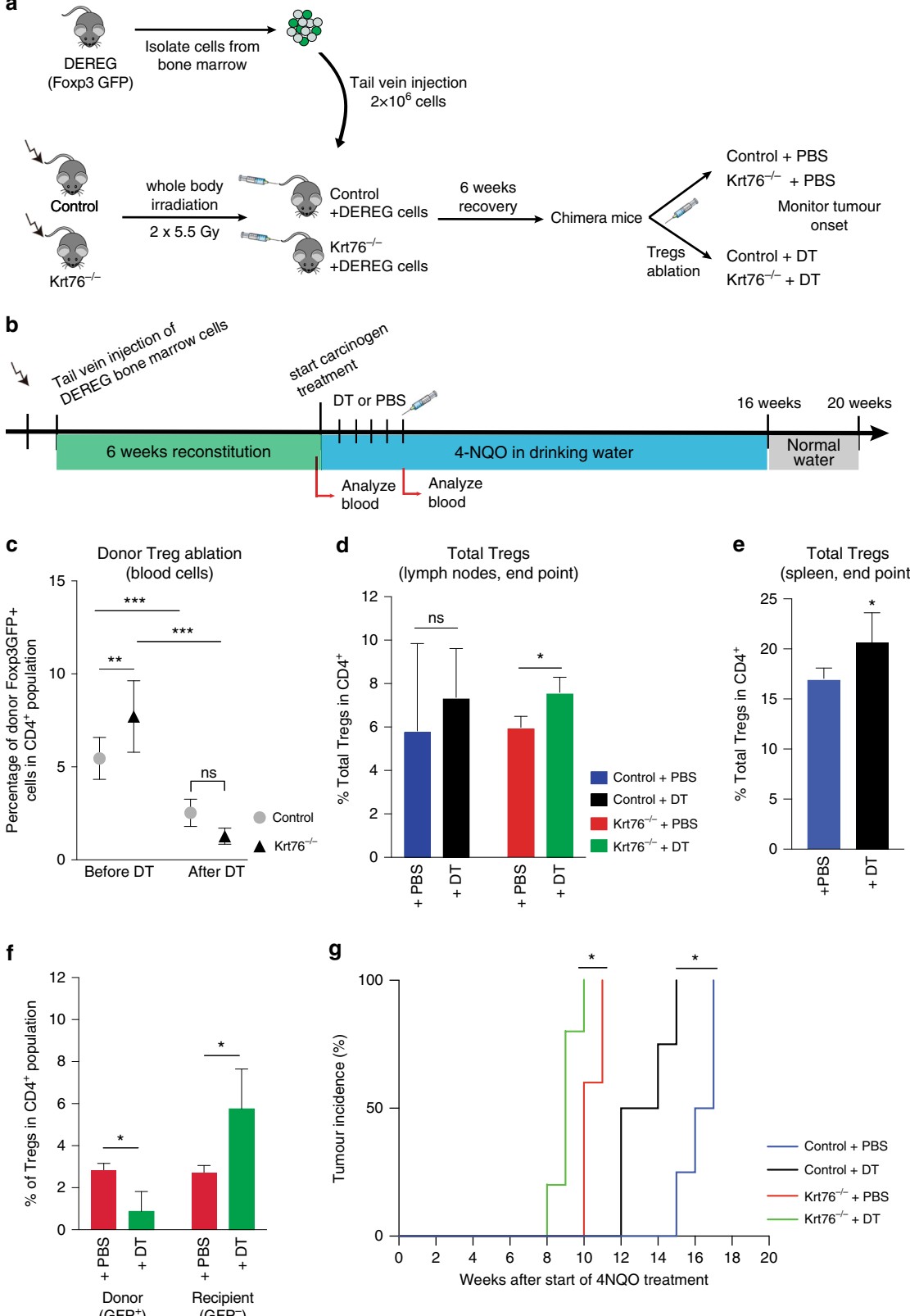

(eBioTY/11.8), CD39 PE (24DM51), CD44 AlexaFluor700 (IM7) (all from eBioscience, used at 1/100 dilution).

Intracellular staining for Foxp3 PE-Cy7 (FJK.165, eBioscience) was performed after fixation with Foxp3 Fixation Buffer and permeabilization with Foxp3 Permeabilization Buffer (eBioscience). 4′,6-Diamidino-2-phenylindole (DAPI, Molecular Probes) was used to exclude dead cells, or Fixable Viability Dye eFluor 455UV (eBioscience) in intracellular staining. Labelled cells were analysed on a BD

LSRFortessa cell analyser. All data were analysed using FlowJo software. The gating strategy for Treg and effector T cell populations is shown in Supplementary Fig. 4.

**Histology**. For frozen sections, tissues were embedded in OCT (optimal cutting temperature compound), sectioned and post-fixed in 4% paraformaldehyde/PBS pH 7.4 for 10 min before staining. X-gal staining on frozen sections was performed

**Fig. 8** 4NQO-induced carcinogenesis in DEREG chimeric mice. **a** Experimental scheme. Recipient mice were irradiated with 5.5 Gy twice and received 2 × 10^6 cells from DEREG mouse BM by tail vein injection. Mice were allowed to recover for 6 weeks before injecting DT to ablate Treg. Control mice were injected with PBS. **b** Schematic of BM transfer and Treg ablation by injecting PBS control or DT for 5 weeks, followed by the 4NQO tumorigenesis protocol. **c** Efficiency of engraftment by Foxp3GFP$^+$ donor Tregs and percentage of donor Treg depletion after DT treatment confirmed by flow cytometric analysis of blood cells. **d, e** Summary of flow cytometric analysis of % total Tregs (CD4$^+$ CD25$^+$ Foxp3$^+$) in total CD4$^+$ cells in lymph nodes (**d**) and spleen (**e**) of control DEREG/Krt76$^{+/-}$ (**d, e**) and DEREG/Krt76$^{-/-}$ (**d**) chimeric mice treated with DT or PBS. ($n = 4$–6 animals/group, mean ± s.e.m., Mann–Whitney test). **f** Summary of flow cytometric analysis of % donor (CD4$^+$ CD25$^+$ Foxp3$^+$ GFP$^+$) and recipient (CD4$^+$ CD25$^+$ Foxp3$^+$ GFP$^-$) Foxp3$^+$ Tregs in total CD4$^+$ lymph node cells from DEREG/Krt76$^{-/-}$ chimera mice treated with DT or PBS ($n = 4$–6 animals/group, mean ± s.e.m., unpaired $t$-test). **g** Tumour incidence in DEREG/Krt76$^{+/+}$ and DEREG/Krt76$^{-/-}$ chimeric mice treated with DT or PBS ($n = 5$–6 animals/group, one-way ANOVA, Mantel-Cox test, $p = 0.013$ for DEREG/Krt76$^{-/-}$ and $p = 0.016$ DEREG/Krt76$^{+/-}$). *$p \leq 0.05$, **$p \leq 0.01$, ***$p \leq 0.001$; ns non-significant

as described[54]. The tissues were sectioned and stained with haematoxylin and eosin (H&E) and with Gram stain (Sigma) by conventional methods. Images were acquired using a Hamamatsu slide scanner and analysed using NanoZoomer software (Hamamatsu).

**Immunofluorescence staining**. Frozen sections were fixed in 2% paraformaldehyde/PBS pH 7.4 and blocked with 10% goat serum, 2% BSA, 0.02% fish skin gelatin and 0.05% TritonX100 (Sigma) in PBS for 1 h at room temperature. Paraffin sections of human OSCC were subjected to heat-mediated antigen retrieval (citrate buffer, pH6) prior to blocking. Primary antibodies were incubated overnight at 4 °C, followed by 1 h incubation at room temperature in secondary antibody.

The following primary antibodies were used: Foxp3 (eBioscience, clone FJK-16s, 1/100, and Abcam, clone 236 A/E7, 1/50), anti-Loricrin, anti-Krt76 (Santa Cruz, clone F-12, 1/100, and Sigma, HPA019656, 1/100), Krt14 (Covance, PRB-155P, 1/1000), B220/CD45R (eBioscience, clone RA3-6B2, 1/100), CD3 (BD Pharmingen, clone 17A2, 1/150), CD45 (BD Pharmingen clone 30-F11, 1/150); and secondary antibodies: anti-goat, anti-mouse and anti-rabbit Alexa Fluor 488, 568 and 633 (Life Technologies, 1/300).

EdU staining was performed with a Click-it EdU imaging kit (Life Technologies) according to the manufacturer's recommendations. DAPI (Life Technologies) was used as a nuclear counterstain. Slides were mounted using ProLong Gold anti-fade reagent (Life Technologies). Images were acquired with a Nikon A1 Upright Confocal microscope. Images were analysed using ICY image analysis software[55].

**Quantification of cell number and epithelial thickness**. Images were quantified using open-source ICY software plug-in Manual Counting[55]. Positively stained cells for CD45, EdU or Foxp3 were counted per length of the epithelium analysed or per stromal area (as delineated in Fig. 3). The total number of nuclei was quantified using DAPI staining. All cell count analyses were performed using sequential sections from well-oriented tongue or stomach blocks in a blind manner on an average of 10 independent fields per animal ($n = 3$–6 animals/genotype/experiment). Representative images from two to three independent experiments with at least three biological replicates per group are shown. All statistical analyses were carried out using Prism 7 (Graph Pad).

**Fluorescence in situ hybridisation with BacUni**. Pieces of tongue and stomach epithelia were collected and sectioned in a sterilised setup. Sections were fixed in 1:1 acetone-to-methanol and incubated with PNA FISH probes at 55 °C. A universal bacteria probe (BacUni; AdvanDx) and probe for *C. dubuliniensis* (AdvanDx) were used[25]. DAPI was used for nuclear counterstaining. Slides were mounted using ProLong Gold anti-fade reagent (Life Technologies). Images and Z-stacks were acquired with a Nikon A1 Upright Confocal microscope. Evaluation of bacterial location was performed on three-dimensional z-stacks using ICY software.

**Epithelial permeability assays**. Whole embryos or dissected tongues were dehydrated through a methanol series (25, 50, 75 and 100% methanol, 60 s per step), rehydrated in PBS, and then stained immediately with 0.1% toluidine blue in water for 10 min, with agitation at room temperature. Samples were briefly washed in PBS and immediately photographed using a Nikon SWZ18 and Nikon DS-Ri2 camera.

Stomach barrier function was assessed using a FITC-labelled dextran method. Briefly, food and water were withdrawn for 4 h and mice were sedated with inhaled isoflurane and orally administrated with the permeability tracer by oral gavage (60 mg/100 g body weight of FITC-labelled dextran, MW 4000; FD4, Sigma-Aldrich). After 30 min, blood was collected by cardiac puncture into Microtainer SST tubes and fluorescence intensity was determined using GloMax® Discover Multimode Microplate Reader (Promega; excitation, 492 nm; emission, 525 nm). FITC-dextran concentrations were determined using a standard curve generated by serial dilution of FITC-dextran.

**Tissue collection, RNA extraction and gene-expression analysis**. Biopsies were homogenised using GENTLEMACS (Miltenyi Biotec) according to the manufacturer's recommendations. Total RNA from homogenised tissues was isolated and purified using the Purelink RNA micro kit (Invitrogen) with on-column DNaseI digestion, according to the manufacturer's instructions. Complementary DNA was generated using SuperScriptIII (Invitrogen). Quantitative real-time PCR reactions were performed with TaqMan Fast Universal PCR Master Mix and Taqman probes (Thermo Fisher Scientific, Supplementary Table 1) on a 7900HT real-time PCR machine (Applied Biosystems) on biological triplicates.

**In vitro T-cell suppression assay**. CD4$^+$ T lymphocytes were purified from splenic cell suspensions using Dynabeads® Untouched™MouseCD4 Cells kits (Life Technologies) followed by CD25$^+$ magnetic-activated cell sorting (MACS) according to the manufacturer's instructions (Miltenyi Biotec). Cell purity was > 96% as determined by flow cytometry. $1 \times 10^6$ cells/ml were resuspended in complete RPMI-1640 medium (Thermo Fisher Scientific) containing L-glutamine (2 mM, Gibco), penicillin/streptomycin (100 U/ml, Gibco), HEPES (1 mM, Gibco), 2-Mercapthoethanol (50 mM, Gibco) and 10% foetal calf serum (Thermo Fisher Scientific). CD4$^+$ CD25- T effector cells (Tresp) were labelled with 5 nM CFSE (Vybrant CFDA SE Cell Tracer kit, Molecular Probes, Life Technologies) for 15 min at 37 °C, resuspended at a concentration of $1 \times 10^6$ cells/ml in complete RPMI-1640 medium and co-cultured with different numbers of CD4$^+$ CD25$^+$ Tregs at a range of dilution ratios, in the presence of $10 \times 10^5$ antigen-presenting cells and monoclonal anti-CD3e mAb (1 μg/ml, clone 145-2c11, BD Pharmingen) in a final volume of 200 μl culture medium in 96-well round-bottom plates. Each sample was tested in duplicate or triplicate. After culture for 72 h at 37 °C in a humidified atmosphere of 5% CO$_2$, Near-IR Live/Dead fixable dead cell stain kit (Thermo-Fisher) was added to exclude dead cells and CFSE dilution was analysed by flow cytometry using a BD LSRFortessa. All data were analysed using FlowJo software. Relative proliferation was calculated as proliferation index without or with Tregs and % suppression calculated.

**Cytokine analysis by cytometric bead array multiplex assays**. Blood was collected by cardiac puncture and left to clot for 2 h at room temperature. Serum was collected after centrifugation for 20 min at 4 °C. Supernatants from the suppression assays were collected and kept at −80 °C prior to analysis. Levels of IFNγ, IL-2, IL-4, IL-6, IL-10 and TNFα were analysed in the serum or in cell culture supernatants using a mouse cytometric bead array Th1/Th2/ Th17 cytokine kit (BD Biosciences) following the manufacturer's instructions. Data were acquired using the BD FACSCanto system (BD Biosciences) and analysed using FlowJo software. All samples were measured in technical duplicates and biological replicates ($n = 4$ for each group).

**Statistical analysis**. All graphs and statistical calculations were generated using Prism7 (GraphPad) software. Statistical significance was computed with the test indicated in each figure legend. The number of experiments and animals analysed are indicated in each figure.

**Data availability**. The data that support the findings of this study are available within the manuscript and its supplementray information or from the authors upon reasonable request.

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

## Acknowledgements

The authors are very grateful to Caetano Reis e Sousa and Neil Rogers for the DEREG mice, Tomasz Zabinski, Inês M. Tomás, Simon Broad, Eamonn Morrison, Matteo Battilochi, Ronan Lyne and BSU staff for assistance with experiments, and Nicholas Wright and Marnix Jansen for advice on histopathology. F.M.W. gratefully acknowledges the financial support of the Wellcome Trust (206439/Z/17/Z), the UK Medical Research Council (MR/PO18823/1) and Cancer Research UK (C219/A23522). J.F.N. is the recipient of a RCUK/UKRI Rutherford Fund fellowship (MR/R024812/1) and gratefully acknowledges funding from the Wellcome Trust (Seed Award in Science, 204394/Z/16/Z) and the European Union (Marie Skolodowska-Curie individual fellowship). G.M.L. acknowledges the financial support of the UK Medical Research Council (MR/M003493/1). Patient tissue samples of oral squamous cell carcinoma were provided by Guy's & St Thomas' Head & Neck Biobank (KHP Cancer Biobank). The authors also acknowledge funding from the Department of Health via the National Institute for Health Research comprehensive Biomedical Research Centre award to Guy's & St Thomas' National

Health Service Foundation Trust in partnership with King's College London and King's College Hospital NHS Foundation Trust.

## Author contributions

I.S. and F.M.W. conceived the project. I.S. led the execution of all experiments, data analysis and manuscript production. J.F.N and G.M.L. contributed expertise on cytokines and immune cell types and J.F.N. performed flow cytometry experiments. Q.P. and G.L. performed and analysed suppression assays. D.C., K.L. and N.P. assisted in performing, quantifying and analysing experiments. P.R.M. contributed to histopathology analysis. F. M.W. participated in the interpretation of the results and manuscript production.

## Additional information

**Competing interests:** F.M.W. is currently on secondment as Executive Chair of the UK Medical Research Council. The remaining authors declare no competing interests.

