## [Peer Review File · Nature Communications]

Reviewers' comments:

Reviewer #1 (Remarks to the Author):

In this study, the authors focus on potential mechanisms by which loss of Krt76 in the context of chemical-induced carcinogenesis within oral and stomach epithelia contributes to increased tumor incidence. The authors first characterize Krt76 expression throughout embryonic development and find Krt76 expression in the tongue, palate, and stomach. Next, the authors observed that Krt76^{-/-} mice develop cysts adjacent to the salivary gland that contain B and T cells, and suggest that the cysts are enlarged submandibular lymph nodes. Lymph nodes and spleens were increased in size relative to littermate controls, and frequencies of B and T cells, including Treg cells, were increased in lymph nodes of Krt76^{-/-} mice. Certain cytokines, including IL-6, IL-10, and TNF α were found at increased levels in the serum, tongue, and squamous stomach of Krt76^{-/-} mice, suggesting increases in local and systemic inflammation. The authors then tested the role of Krt76 upon treatment of mice with the synthetic carcinogen 4NQO, which leads to hyperplasia in the oral epithelium and the development of oral squamous cell carcinoma by 22 weeks. Krt76 was frequently down regulated in the lesions, which the authors suggest to be analogous to Krt76 loss in human cancer. Krt76^{-/-} developed tumors earlier, and also developed tumors in the squamous stomach at an increased frequency relative to wild-type mice. The authors found that 4NQO-treated Krt76^{-/-} mice exhibited higher increases in serum levels of IFN γ , IL-4, IL-6, IL-10, and TNF α than control mice, as well as increased immune cell infiltrate and higher numbers of Treg cells within hyperplastic and dysplastic lesions. The authors suggest that increased levels of IL-18 and IL-33 mRNA levels in the tongue and squamous stomach early on after 4NQO treatment may be responsible for the increased Treg cell numbers. Finally, the authors report that in human OSCC, increased Foxp3⁺ cells are found in Krt76-deficient regions.

The study demonstrates that loss of Krt76 is associated with accelerated development of lesions in oral and squamous stomach epithelia and with increased levels of immune infiltration. While Krt76-deficient mice have previously been shown to be more susceptible to oral hyperplasia (Ambatipudi et al. PLOS ONE 2014), the increased levels of inflammatory cytokines, lymphadenopathy and splenomegaly, and increased immune cell infiltrate in the lesions are novel observations. Nevertheless, the study remains mainly descriptive and does not establish a causal mechanistic link between Krt76 loss and immunomodulation.

Major Points:

1. While the study shows increased Treg cell numbers within dysplastic lesions formed in Krt76^{-/-} mice, it is not clear whether activated effector T cell populations are concomitantly increased. It is likely that breach in the epithelial barrier integrity results in heightened immune activation and activation of negative regulatory mechanisms including increase in Treg cell numbers. Is the observed activation of the immune cells associated with increased translocation of commensal microorganisms?
2. It remains unknown whether the Treg cells or effector T cells are contributing to the heightened tumor growth. The immune status of tumors has not been assessed. In this regard, in a genetic model of skin squamous carcinoma Doug Hanahan's group reported that absence of CD4 T cells sharply decreased tumor incidence (J Exp Med, 2003).
3. Several inflammatory cytokines are found at increased levels in serum, lymphoid cysts, and tumor tissue of Krt76-deficient mice. Chronic inflammation is strongly associated with the development of certain tumors such as colorectal cancer; in colorectal cancer it has been suggested that the presence of Treg cells may actually be associated with diminished carcinogenesis through a decrease in tumor-promoting inflammation.
4. It is possible that the accelerated tumor development observed in Krt76-deficient mice may be

driven by chronic inflammation caused by increased levels of inflammatory cytokines rather than by increased attraction of Treg cells. Considering that Krt76 deficient mice have considerable phenotype, interpretations of the observed changes in carcinogenesis are wide open.

Minor Points:

1. The numbers of splenocytes in control animals ($\sim 1.5 \times 10^7$) shown in Figure 1 are unusually low. In healthy 6-8 wk old mice, splenocyte numbers are $\sim 5 \times 10^7$ - 10^8 cells. What's the reason for such low numbers?
2. In Figures 3b and 3c, total T cell numbers need to be shown. In addition, it is not obvious whether the percentage listed is the percentage of CD4+ T cells that are Foxp3+ or the percentage of total cells that are CD4+Foxp3+.
3. In Figure 4g, the y-axis might better be displayed as percentage of animals.
4. In Figure 5b, the immunofluorescence images purporting to show increased accumulation of CD45+ cells near the abnormal epithelium in Krt76-/- mice are unclear—it's difficult to distinguish individual CD45+ cells. These data do not obviously support the idea of localized accumulation.
5. Similarly, in Figures 5g-h, the identification of individual Foxp3+ cells is difficult and the non-specific staining near the epithelial wall is more obvious than the nuclear foci to which the arrows point. Either higher quality or higher contrast images would help, as would flow cytometric staining of tissue containing abnormal growth to survey for Foxp3+ CD4+ T cells.

Reviewer #2 (Remarks to the Author):

In this manuscript by Sequeira et al., the authors identify an interesting role for the cytoskeletal structural protein Keratin 76 in keeping the immune system in check and preventing oral and gastric cancer. Using whole-body Krt76-deficient mice, the authors convincingly demonstrate that these mice experience systemic inflammation and are more prone to tumorigenesis. However, given the prior literature, the manuscript as it currently stands falls short on several aspects that need to be strengthened to move forward towards publication:

- 1) The authors mention that the premise of this study is the striking downregulation of K76 in human OSCC, and aim to find a mechanistic connection. However, since K76 is a marker of differentiated cell lineages (barrier function), the reviewer would argue that loss of K76 in OSCC could just reflect an expansion of the undifferentiated state seen in tumors. If the authors wish to connect loss of K76 in human OSCC as causal drivers of tumorigenesis, they could study the TCGA database of human cancer and look for common mutations in Krt76.

The Krt76-deficient mouse model, when challenged with chemical agents, no doubt has a higher propensity to develop tumors. But the reviewer is not convinced that in human SCC, the loss of K76 is involved in the early steps of tumorigenesis. The authors even mention that in their mouse model, "some lesions had lost Krt76 expression whereas others retained it", suggesting that loss of Krt76 is not directly linked to tumorigenesis.

- 2) DiTommaso et al. PLOS genetics, 2014 had previously described that Krt76-deficient mice display barrier defects and inflammation, and K76 stabilizes epithelial tight junction by interacting with tight junction components such as Claudin1. The present study expands this concept to the oral epithelium and stomach. The most novel angle is the tumorigenesis studies. However, to be convincing, the tumorigenesis study needs more thorough analyses. According to DiTommaso et. al., Krt76-/- mice show tight-junction barrier defects, and this might increase the penetration rate of mutagen (4NQO) in Krt76-/- mice. Thus, it is possible that the increased cancer incidence in Krt76-/- results from higher local concentration of 4NQO in the basal progenitor cells. To exclude this possibility, the authors should be able to perform a couple of extensive experiments

addressing the epithelial integrity and the penetration of 4NQO in Krt76^{-/-} mice. The authors could also use genetic tumor models (ex. Ras/p53 mutations or manipulation of other oncogenes and tumor suppressor genes).

3) Chronic inflammation has long been linked to tumorigenesis, and several mechanisms are studied and proposed. The novel and intriguing finding of the manuscript would be the link between K76 loss and systemic inflammation. However, the authors did not address the question of how K76 deficiency induces increases of local and systemic inflammation. Does it result from epithelial barrier defects? Does K76 play a role in transcriptional regulation of inflammatory genes in the nucleus? The manuscript could be strengthened by providing more detailed molecular insights as to how K76 keeps the immune system in check. The current results suggest that it is the loss of barrier function, rather than a specific role of K76, which triggers the immune response, but this needs to be dissected further.

4) In Figures 2h and 2i, images of control/WT lymph nodes are missing. This is critical to assess the presence/abundance of B cells (B220 marker) and T cells in healthy mice. In addition, the authors should provide the immunostaining of K76 in lymph nodes, instead of just mentioning this without showing data.

Related to that, the Flow Cytometry plots in 2j don't show striking differences on the level of T cells between WT and Krt76 KO mice, both in thymus and lymph nodes. This emphasizes why adding the immunofluorescence images in 2h and 2i of WT lymph nodes becomes important.

5) The quantifications in Figure 3h: Based on the immunofluorescence images of skin (Figure 3g), it appears as if there are a lot more CD45⁺ cells in the KO skin than reflected in the quantification. The reviewer is also confused by the axis labeling in 3h: e.g. stomach in WT: 0.01 cells per mm²? If the label is correct, and accurately reflects calculations, the reviewer would still suggest to illustrate this in a more intuitive manner.

Minor points:

1) Images: The expression pattern of K76 is sometimes not apparent from the images shown. E.g. Fig 1c, zooming in would help (oral cavity), stomach: make K76 (green) brighter. It's also unclear where to look for the Xgal staining at E17.5 (especially stomach). The authors could consider adding arrows?

2) Figure labels: It is sometimes unclear what the figures are showing, without looking carefully at the legends. The authors could add a title to the graphs, e.g. Figure 2e, 2f, 2k, 2o, 2r, 4b

3) Typo: Reference #6. "Nature Publishing Group" -> "Nature Genetics"

Response to Reviewers' comments:

Reviewer #1 (Remarks to the Author):

In this study, the authors focus on potential mechanisms by which loss of Krt76 in the context of chemical-induced carcinogenesis within oral and stomach epithelia contributes to increased tumor incidence. The authors first characterize Krt76 expression throughout embryonic development and find Krt76 expression in the tongue, palate, and stomach. Next, the authors observed that Krt76^{-/-} mice develop cysts adjacent to the salivary gland that contain B and T cells, and suggest that the cysts are enlarged submandibular lymph nodes. Lymph nodes and spleens were increased in size relative to littermate controls, and frequencies of B and T cells, including Treg cells, were increased in lymph nodes of Krt76^{-/-} mice. Certain cytokines, including IL-6, IL-10, and TNF α were found at increased levels in the serum, tongue, and squamous stomach of Krt76^{-/-} mice, suggesting increases in local and systemic inflammation. The authors then tested the role of Krt76 upon treatment of mice with the synthetic carcinogen 4NQO, which leads to hyperplasia in the oral epithelium and the development of oral squamous cell carcinoma by 22 weeks. Krt76 was frequently down regulated in the lesions, which the authors suggest to be analogous to Krt76 loss in human cancer. Krt76^{-/-} developed tumors earlier, and also developed tumors in the squamous stomach at an increased frequency relative to wild-type mice. The authors found that 4NQO-treated Krt76^{-/-} mice exhibited higher increases in serum levels of IFN γ , IL-4, IL-6, IL-10, and TNF α than control mice, as well as increased immune cell infiltrate and higher numbers of Treg cells within hyperplastic and dysplastic lesions. The authors suggest that increased levels of IL-18 and IL-33 mRNA levels in the tongue and squamous stomach early on after 4NQO treatment may be responsible for the increased Treg cell numbers. Finally, the authors report that in human OSCC, increased Foxp3⁺ cells are found in Krt76-deficient regions.

The study demonstrates that loss of Krt76 is associated with accelerated development of lesions in oral and squamous stomach epithelia and with increased levels of immune infiltration. While Krt76-deficient mice have previously been shown to be more susceptible to oral hyperplasia (Ambatipudi et al. PLOS ONE 2014), the increased levels of inflammatory cytokines, lymphadenopathy and splenomegaly, and increased immune cell infiltrate in the lesions are novel observations. Nevertheless, the study remains mainly descriptive and does not establish a causal mechanistic link between Krt76 loss and immunomodulation.

Major Points:

1. While the study shows increased Treg cell numbers within dysplastic lesions formed in Krt76^{-/-} mice, it is not clear whether activated effector T cell populations are concomitantly increased. It is likely that breach in the epithelial barrier integrity results in heightened

immune activation and activation of negative regulatory mechanisms including increase in Treg cell numbers. Is the observed activation of the immune cells associated with increased translocation of commensal microorganisms?

As suggested by the reviewer, we have now examined effector T cells in control and knockout mice (see new Figure 3a). We find that in Krt76 knockout mice there is an increase in effector T cells (CD4+ CD44+ CD62low) in the circulation and peripheral organs.

Regarding the lesions, 4NQO treatment results in a decrease in effector T cell populations (CD4+ CD44+) in Krt76-/- tongue and stomach lesions (new Figure 6i). This suggests that the suppressor function of Tregs is greater in Krt76-/- lesions. We confirmed this by performing in vitro suppression assays (new Figure 4). We observed higher suppressive capacity of Krt76-/- Tregs compared to controls, together with increased expression of the functional markers CD39 and CD73 (new Figure 4e, f).

To examine whether there is increased translocation of commensal microorganisms across Krt76-/- epithelia that could lead to an activation of immune cells, we measured the bacterial load (new Supplementary Figure 3) by Gram staining of the tongue and stomach. We also performed whole-mount fluorescence in situ hybridization (FISH) with a universal bacterial probe (BacUni) or a C. dubuliniensis probe as a negative control (cf. Natsuga et al., 2016, now cited). Bacterial penetration was largely confined to the cornified layers of the tongue filiform papillae and forestomach in both Krt76-/- and control mice. The labeling demonstrated that there was no difference in bacterial load and penetration between Krt76-/- and control mice.

2. It remains unknown whether the Treg cells or effector T cells are contributing to the heightened tumor growth. The immune status of tumors has not been assessed. In this regard, in a genetic model of skin squamous carcinoma Doug Hanahan's group reported that absence of CD4 T cells sharply decreased tumor incidence (J Exp Med, 2003).

In Figure 6, we quantified the immune status of tumours and observed an increase in CD45+ cells and Foxp3+ Tregs. We have now analysed the total number of CD4+ cells and effector T cells in tumours (new Figure 6i). We observe a significant increase in CD4+ cells and a decrease in effector T cells (CD4+ CD44+ CD62L low) in the lesions, consistent with the less proliferative status of these cells (Figure 4c). Thus the increased tumour incidence in Krt76-/- mice is correlated with an increase in CD4+ cells concomitant with an increase in Tregs and decrease in effector T cells.

As regards to the immune profiling and additional mechanistic studies, we believe that the inclusion of data on effector T cells (see point 1) is helpful. We have also performed in vitro assays to assess the activity of CD25+CD4+ Tregs from control and Krt76-/-

mice in suppressing proliferation of CD4+CD25- responder T cells. We co-cultured CD4+CD25- responder T cells from control or Krt76-/- mice with Tregs from control or Krt76-/- mice, and stimulated them with monoclonal anti-CD3 antibody in the presence of antigen-presenting cells. When compared with Tregs from control mice, those from Krt76-/- mice exhibited a substantially increased suppressive function independent of whether the responder T cells were from control or Krt76-/- mice (new Figure 4). We also observed that responder T cells from Krt76-/- mice proliferated less, even in the absence of Tregs.

The increased suppression capability of CD25+CD4+ Tregs from K76-/- mice correlates with increased capacity to inhibit pro-inflammatory cytokine production (IFN gamma and IL17) by CD4+ responder T cells (new Figure 4). Finally, immunophenotyping of the Tregs demonstrates an upregulation of functional markers: we have observed higher CD39 and CD73 expression and an increased percentage of CD39+ and CD73+ cells within the Treg population.

Our new experiments suggest that in Krt76-/- mice not only are the responder T cells less effective in their anti-tumour response, but also the Tregs are more suppressive.

3. Several inflammatory cytokines are found at increased levels in serum, lymphoid cysts, and tumor tissue of Krt76-deficient mice. Chronic inflammation is strongly associated with the development of certain tumors such as colorectal cancer; in colorectal cancer it has been suggested that the presence of Treg cells may actually be associated with diminished carcinogenesis through a decrease in tumor-promoting inflammation.

We agree with these comments. However, although there is an increase in the CD45+ cells and in some inflammatory cytokines, there is no evidence of hyperproliferation or development of spontaneous tumours in the tongue or stomach epithelia of Krt76-/- mice (Figure 2a and b).

4. It is possible that the accelerated tumor development observed in Krt76-deficient mice may be driven by chronic inflammation caused by increased levels of inflammatory cytokines rather than by increased attraction of Treg cells. Considering that Krt76 deficient mice have considerable phenotype, interpretations of the observed changes in carcinogenesis are wide open.

We believe that the new experiments described above implicate Tregs rather than chronic inflammation. Furthermore, we have directly addressed the role of the Tregs by creating bone marrow chimeras with DEREK (Foxp3 GFP DTR) mice and performing 4NQO carcinogenesis. We chose a protocol that would give a sustained reduction in

Tregs and achieved a 50% decrease in donor Tregs over a period of 5 weeks (new Figure 8). This led to a further increase in tumour formation, which correlated with an increase in the total number of Tregs through enhanced production of Tregs by the recipient mice.

Minor Points:

1. The numbers of splenocytes in control animals ($\sim 1.5 \times 10^7$) shown in Figure 1 are unusually low. In healthy 6-8 wk old mice, splenocyte numbers are $\sim 5 \times 10^7$ - 10^8 cells. What's the reason for such low numbers?

We have examined additional mice and counted spleen cells manually (with a cytometer) and with a cell counter (Scepter™ 2.0 Cell Counter). We can confirm that the original values were correct. According to the literature they correspond to the normal range in the number of splenocytes after RBC lysis in healthy 3-month old C57 mice (0.9×10^7 ; cells/spleen in Pinchuk and Filipov, 2008).

2. In Figures 3b and 3c, total T cell numbers need to be shown. In addition, it is not obvious whether the percentage listed is the percentage of CD4+ T cells that are Foxp3+ or the percentage of total cells that are CD4+Foxp3+.

We have now specified in the Figure legend that the % Tregs in Figure 3c (now Figure 3b) corresponds to the % of CD4+ CD25+Foxp3+ cells within the total T cell population (TCRb+ CD3+ CD4+). New Supplementary Figure 4 shows the gating strategy, complying with the journal's policy on flow cytometry reporting.

3. In Figure 4g, the y-axis might better be displayed as percentage of animals.

The y-axis has now been changed.

4. In Figure 5b, the immunofluorescence images purporting to show increased accumulation of CD45+ cells near the abnormal epithelium in Krt76-/- mice are unclear—it's difficult to distinguish individual CD45+ cells. These data do not obviously support the idea of localized accumulation.

We believe that the higher power inserts are clear, but we have now increased the magnification of these inserts to distinguish individual CD45+ cells (new Figure 6b). We

have also performed flow cytometry of CD45+ immune cells from treated lesions, which supports the quantification based on histology (see new Figure 6i).

5. Similarly, in Figures 5g-h, the identification of individual Foxp3+ cells is difficult and the non-specific staining near the epithelial wall is more obvious than the nuclear foci to which the arrows point. Either higher quality or higher contrast images would help, as would flow cytometric staining of tissue containing abnormal growth to survey for Foxp3+ CD4+ T cells.

We have now increased the quality and resolution of the images to facilitate identification of individual Foxp3+ cells (new Figure 6e, f, g). We have also performed flow cytometry of Foxp3+ CD4+ Treg cells from control and Krt76-/- lesions, supporting the quantification observed by histology (see new Figure 6i).

Reviewer #2 (Remarks to the Author):

In this manuscript by Sequeira et al., the authors identify an interesting role for the cytoskeletal structural protein Keratin 76 in keeping the immune system in check and preventing oral and gastric cancer. Using whole-body Krt76-deficient mice, the authors convincingly demonstrate that these mice experience systemic inflammation and are more prone to tumorigenesis. However, given the prior literature, the manuscript as it currently stands falls short on several aspects that need to be strengthened to move forward towards publication:

1) The authors mention that the premise of this study is the striking downregulation of K76 in human OSCC, and aim to find a mechanistic connection. However, since K76 is a marker of differentiated cell lineages (barrier function), the reviewer would argue that loss of K76 in OSCC could just reflect an expansion of the undifferentiated state seen in tumors. If the authors wish to connect loss of K76 in human OSCC as causal drivers of tumorigenesis, they could study the TCGA database of human cancer and look for common mutations in Krt76.

To explore this interesting suggestion, we labelled mouse and human OSCC with antibodies to the differentiation marker Involucrin. As shown in the new Figure 7, we observed expression of Involucrin even when Krt76 expression was lost. We have, as suggested, examined the TCGA database and now refer to the fact that Krt76 is not mutated in OSCC.

The Krt76-deficient mouse model, when challenged with chemical agents, no doubt has a higher propensity to develop tumors. But the reviewer is not convinced that in human SCC, the loss of K76 is involved in the early steps of tumorigenesis. The authors even mention that in their mouse model, “some lesions had lost Krt76 expression whereas others retained it”, suggesting that loss of Krt76 is not directly linked to tumorigenesis.

We agree that there is no evidence for a causal link, and we have checked the text to make sure that this is not implied. We do, however, provide mechanistic evidence to support the conclusion of poorer prognosis of human OSCC lacking Krt76 is due to immune dysfunction.

2) DiTommaso et al. PLOS genetics, 2014 had previously described that Krt76-deficient mice display barrier defects and inflammation, and K76 stabilizes epithelial tight junction by interacting with tight junction components such as Claudin1. The present study expands this concept to the oral epithelium and stomach. The most novel angle is the tumorigenesis studies. However, to be convincing, the tumorigenesis study needs more thorough analyses. According to DiTommaso et. al., Krt76^{-/-} mice show tight-junction barrier defects, and this might increase the penetration rate of mutagen (4NQO) in Krt76^{-/-} mice. Thus, it is possible that the increased cancer incidence in Krt76^{-/-} results from higher local concentration of 4NQO in the basal progenitor cells. To exclude this possibility, the authors should be able to perform a couple of extensive experiments addressing the epithelial integrity and the penetration of 4NQO in Krt76^{-/-} mice. The authors could also use genetic tumor models (ex. Ras/p53 mutations or manipulation of other oncogenes and tumor suppressor genes).

As a result of this helpful suggestion, we conducted penetration assays in tongue, skin and stomach (see Supplementary Fig. 3). We performed Toluidine Blue labelling of mouse embryos and found that the timing and extent of barrier formation in the tongue was not altered by lack of Krt76 (Supplementary Fig. 3b), but it was delayed in the skin (new Supplementary Fig. 3a) as suggested by DiTommaso et al. Moreover, stomach permeability was measured by determining the concentration of FITC-dextran in blood serum and there were no significant changes between control and Krt76^{-/-} mice (new Supplementary Fig. 3c). We also analysed the mRNA levels of tight-junction proteins such as Claudins and found that they did not differ significantly in tongue or stomach epithelia from control and Krt76^{-/-} mice, although – confirming the findings of DiTommaso et al. – Claudin1, Claudin3 and Claudin 7 expression were reduced in Krt76^{-/-} skin (new Supplementary Fig. 3f).

In the skin of Krt76^{-/-} mice there is suprabasal expression of Krt14 and an increase in the number of suprabasal layers expressing the differentiation markers Loricrin, Filaggrin and Krt10 (Liakath-Ali et al., 2014). However, in the tongue and stomach epithelia expression of Krt14 is confined to the basal layer and there are no changes in the thickness of the suprabasal layers expressing the differentiation markers Loricrin and Filaggrin (new Supplementary Fig. 3d-e). Therefore, we find that loss of Krt76 does not

compromise the integrity and barrier properties of these tissues.

3) Chronic inflammation has long been linked to tumorigenesis, and several mechanisms are studied and proposed. The novel and intriguing finding of the manuscript would be the link between K76 loss and systemic inflammation. However, the authors did not address the question of how K76 deficiency induces increases of local and systemic inflammation. Does it result from epithelial barrier defects?

Does K76 play a role in transcriptional regulation of inflammatory genes in the nucleus? The manuscript could be strengthened by providing more detailed molecular insights as to how K76 keeps the immune system in check. The current results suggest that it is the loss of barrier function, rather than a specific role of K76, which triggers the immune response, but this needs to be dissected further.

As addressed in point 2, the Krt76^{-/-} tongue and stomach epithelia do not have barrier defects (Supplementary Fig. 3). Please see response to Reviewer 1 for the experiments we have performed to look at the link with inflammation, Tregs and effector T cells.

Although Krt17 is found in the nucleus (DePianto et al. 2010, Hobbs et al. 2015) we have failed to detect nuclear Krt76 by antibody labelling. In addition, according to the most accurate tools to predict NLS (NLS mapper and PredictNLS_DB, in Cokol et al., EMBO reports, 2000), Krt76 is not predicted to have any monopartite or bipartite NLS.

4) In Figures 2h and 2i, images of control/WT lymph nodes are missing. This is critical to assess the presence/abundance of B cells (B220 marker) and T cells in healthy mice. In addition, the authors should provide the immunostaining of K76 in lymph nodes, instead of just mentioning this without showing data.

Related to that, the Flow Cytometry plots in 2j don't show striking differences on the level of T cells between WT and Krt76 KO mice, both in thymus and lymph nodes. This emphasizes why adding the immunofluorescence images in 2h and 2i of WT lymph nodes becomes important.

We have now added the immunostaining requested by the reviewer: B and T cell markers in control lymph nodes (Figure 2h and i) and Krt76 in lymph nodes (Supplementary Fig. 1). Although the total number of total CD4⁺ T cells is not significantly altered in thymus and lymph nodes, the proportion and functionality of Tregs and effector T cell subpopulations are altered in Krt76^{-/-} mice, as explained above (Figure 3a, b and new Figure 4).

5) The quantifications in Figure 3h: Based on the immunofluorescence images of skin (Figure 3g), it appears as if there are a lot more CD45+ cells in the KO skin than reflected in the quantification. The reviewer is also confused by the axis labeling in 3h: e.g. stomach in WT: 0.01 cells per mm²? If the label is correct, and accurately reflects calculations, the reviewer would still suggest to illustrate this in a more intuitive manner.

We have re-quantified the CD45+ cells in immunostained sections and corrected the label (new Figure 3f, g). This confirms the increased number of CD45+ cells in the skin, tongue and stomach epithelia. Using ICY image analysis software, the number of CD45+ cells was quantified within the tissue stroma area (in mm²). To make the quantification more intuitive, we have now delineated the region quantified (new Figure 3f).

Minor points:

1) Images: The expression pattern of K76 is sometimes not apparent from the images shown. E.g. Fig 1c, zooming in would help (oral cavity), stomach: make K76 (green) brighter. It's also unclear where to look for the Xgal staining at E17.5 (especially stomach). The authors could consider adding arrows?

This has now been done.

2) Figure labels: It is sometimes unclear what the figures are showing, without looking carefully at the legends. The authors could add a title to the graphs, e.g. Figure 2e, 2f, 2k, 2o, 2r, 4b

We have included titles to the graphs, as suggested.

3) Typo: Reference #6. "Nature Publishing Group" -> "Nature Genetics".

This has now been done.

REVIEWERS' COMMENTS:

Reviewer #1 (Remarks to the Author):

Editorial Note: Reviewer#1 expresses his satisfaction with the authors' revisions in a confidential note to the editor.

Reviewer #2 (Remarks to the Author):

Prior published work by Ambatipudi et al. (2013) reported a strong correlation between oral cancers and low Krt76. In the current study, the authors analyze Krt76 null mice and demonstrate that this link to OSCC susceptibility is causative and not merely correlative. DiTommaso et al. (2014) previously showed that in skin, genetic loss of Krt76 causes inflammation and hyperproliferation by impairing the barrier function. The important question is how mechanistically loss of this keratin causes inflammation.

The focus of this current study is on the "role of epithelial cells in modulating carcinogenesis via communication with cells of the immune system." In the revised manuscript, the authors have now performed additional experiments to show that loss of Krt76 does not impair barrier function in oral and gastric epithelia. This addition was important to distinguish their study from DiTommaso et al. (2014), who showed that genetic loss of Krt76 causes skin inflammation and hyperproliferation by impairing the barrier function. The new finding now makes it clearer that there must be some other explanation.

The authors now also clarify that Krt76 is not commonly mutated in human OSCC, and further rule out a transcriptional role of Krt76 in the regulation of inflammatory genes. They also add more in-depth characterization of the immune phenotype and the cytokines released. Overall, the authors have provided a compelling case that the oral epithelial cells do communicate with cells of the immune system, but if this were the only point, it would not be a novel one. If the authors can provide more insight into how loss of Krt76 is causally linked to suppression of regulatory T cells to result in inflammation and cancer, this could be exciting and important. This critical point remains correlative.

REVIEWERS' COMMENTS:

Reviewer #1 (Remarks to the Author):

Editorial Note: Reviewer#1 expresses his satisfaction with the authors' revisions in a confidential note to the editor.

Reviewer #2 (Remarks to the Author):

Prior published work by Ambatipudi et al. (2013) reported a strong correlation between oral cancers and low Krt76. In the current study, the authors analyze Krt76 null mice and demonstrate that this link to OSCC susceptibility is causative and not merely correlative.

DiTommaso et al. (2014) previously showed that in skin, genetic loss of Krt76 causes inflammation and hyperproliferation by impairing the barrier function. The important question is how mechanistically loss of this keratin causes inflammation.

The focus of this current study is on the “role of epithelial cells in modulating carcinogenesis via communication with cells of the immune system.” In the revised manuscript, the authors have now performed additional experiments to show that loss of Krt76 does not impair barrier function in oral and gastric epithelia. This addition was important to distinguish their study from DiTommaso et al. (2014), who showed that genetic loss of Krt76 causes skin inflammation and hyperproliferation by impairing the barrier function. The new finding now makes it clearer that there must be some other explanation.

The authors now also clarify that Krt76 is not commonly mutated in human OSCC, and further rule out a transcriptional role of Krt76 in the regulation of inflammatory genes. They also add more in-depth characterization of the immune phenotype and the cytokines released. Overall, the authors have provided a compelling case that the oral epithelial cells do communicate with cells of the immune system, but if this were the only point, it would not be a novel one. If the authors can provide more insight into how loss of Krt76 is causally linked to suppression of regulatory T cells to result in inflammation and cancer, this could be exciting and important. This critical point remains correlative.

In addition to the point that oral epithelial cells communicate with cells of the immune cells our study shows that loss of Krt76 increases cancer susceptibility, rather than reflecting the differentiated status of tumours. We feel that both points come over clearly in the text and have not, therefore, made any further changes.